# A multi-layer physically-based snowpack model simulating direct and indirect radiative impacts of light-absorbing impurities in snow

Francois Tuzet[1,2], Marie Dumont[1], Matthieu Lafaysse[1], Ghislain Picard[2], Laurent Arnaud[2], Didier Voisin[2], Yves Lejeune[1], Luc Charrois[1], Pierre Nabat[3], and Samuel Morin[1]

[1]Meteo-France - CNRS, CNRM UMR 3589, Centre d'Etudes de la Neige, Grenoble , France

[2]UGA,CNRS, Institut des Geosciences de l'Environnement (IGE) UMR 5001, Grenoble, France

[3]Meteo-France - CNRS, CNRM UMR 3589, GMGEC/MOSCA, Toulouse , France

*Correspondence to:* Francois Tuzet (francois.tuzet@meteo.fr)

**Abstract.**

Light-absorbing impurities decrease snow albedo, increasing the amount of solar energy absorbed by the snowpack. Its most intuitive and direct impact is to accelerate snow melt. Enhanced energy absorption in snow also modifies snow metamorphism, which can indirectly drive further variations of snow albedo in the near-infrared part of the solar spectrum because of the
evolution of the near-surface snow microstructure. New capabilities have been implemented in the detailed snowpack model SURFEX/ISBA-Crocus (referred to as Crocus) to account for impurities deposition and evolution within the snowpack and their direct and indirect impacts. Once deposited, the model computes impurities mass evolution until snow melts out, accounting for scavenging by melt water. Taking benefits of the recent inclusion of the spectral radiative transfer model TARTES in Crocus, the model explicitly represents the radiative impacts of light-absorbing impurities in snow.

The model was evaluated at the Col de Porte experimental site (French Alps) during the 2013-2014 snow season, against in-situ standard snow measurements and spectral albedo measurements. In-situ meteorological measurements were used to drive the snowpack model, except for aerosol deposition fluxes. Black carbon and dust deposition fluxes used to drive the model were extracted from simulations of the atmospheric model ALADIN-Climate. The model simulates snowpack evolution reasonably, providing similar performances to our reference Crocus version in term of snow depth, snow water equivalent, near-surface
specific surface area and shortwave albedo. Since the reference empirical albedo scheme was calibrated at Col de Porte, improvements were not expected to be significant in this study. We show that the deposition fluxes from the ALADIN-Climate model provide a reasonable estimate of the amount of light-absorbing impurities deposited on the snowpack except for extreme deposition events which are greatly underestimated. For this particular season, the simulated melt-out date advances by 6 to 9 days due to the presence of light-absorbing impurities. The model makes it possible to apportion the relative importance of
direct and indirect impacts of light-absorbing impurities on energy absorption in snow. For the snow season considered, the direct impact in the visible part of the solar spectrum accounts for 85% of the total impact, while the indirect impact related to accelerated snow metamorphism decreasing near-surface specific surface area and thus decreasing near-infrared albedo,

accounts for 15% of the total impact. Our model results demonstrate that these relative proportions vary with time during the season, with potentially significant impacts for snow melt and avalanche prediction.

# 1 Introduction

Light-absorbing impurities (LAIs) in snow increase the absorption of solar radiation in the visible range, warming up the snowpack and accelerating snow melt (e.g., Warren and Wiscombe, 1980; Jacobson, 2004). Snow albedo can be affected by a wide variety of impurities such as mineral dust (Painter et al., 2010), black carbon (BC) from combustion sources (Flanner et al., 2007), volcanic ash (Conway et al., 1996), soil organics (Takeuchi, 2002), algae, and other biological organisms and constituents (Cook et al., 2017). The concentrations of these impurities in snow are determined by their mixing ratio in precipitation (wet deposition), the amount deposited to the surface through dry deposition and by impurity redistribution in the snowpack via post-depositional processes such as wind-driven drifting, wind pumping, snow sublimation, and scavenging during snow melt which contributes to decrease the surface concentration of LAI at melt time (Doherty et al., 2013). Besides impurities, which operate mostly in the visible part of the solar spectrum, the physical properties of the snow microstructure also influence snow albedo and light penetration in snow, in particular in the near-infrared. This concerns in particular density and specific surface area (SSA) (Domine et al., 2006). Therefore, addressing the impact of light absorbing impurities in snow must also take into account physical snow properties. Indeed, the radiative impact of LAIs on snow can be separated in two parts, direct and indirect impacts (Painter et al., 2007). LAIs in snow accelerates snow melt through albedo feedbacks: the darkening of the snow surface reduces albedo in the visible range (direct impact). This leads to an acceleration of the metamorphism which further reduces albedo by accelerating near-surface SSA decrease (indirect impacts). This induces at least two positive snow albedo feedbacks. First, snow albedo in the near-infrared decreases with SSA (even in absence of LAIs due to a decrease in the ratio between scattering and absorption coefficients; Warren, 1982). Secondly, for a given LAI concentration in snow, LAI radiative forcing increases as SSA decreases (Doherty et al., 2013).

The LAI content in snow has been the subject of numerous measurements. For instance, Carmagnola et al. (2013), Aoki et al. (2014) and Polashenski et al. (2015) gathered information on the snow LAI content (insoluble soot and dust) over the Greenland Ice Sheet. Doherty et al. (2010) also focused on the radiative impact of LAIs on arctic snow showing in particular that non-BC constituents (e.g. organic carbon) are responsible for up to 50% of absorption by LAIs in the Arctic. Bisiaux et al. (2012) presents a review of BC deposition in Antarctic over the last century derived from ice core analysis. LAI content in seasonal snowpacks has also been the subject of several studies. Painter et al. (2013b) and Skiles et al. (2015) pointed out that in the upper Colorado basin, dust strongly affects snow radiative forcing and can advance total melt-out by up to 50 days. Sterle et al. (2013) showed how the impurity content evolves with respect to snow metamorphism and melt, notably that the accumulation of BC and dust on the top of the snowpack at the end of the season plays an important role on the radiative forcing of Sierra Nevada's spring snowpack. In the European Alps, the two types of LAIs suspected to have the most significant influence on snowpack evolution are BC and mineral dust (Di Mauro et al., 2015). Table 2 in Libois et al. (2013) summarized measurements of BC concentration in snow in different sites in the Alps, highlighting that BC is present in snow even in sites remote from the main BC sources. Painter et al. (2013a) even estimated BC to be one of the causes of the end of the Little Ice Age in the Alps. Mineral dust deposition are also frequently observed across the European mountain ranges, giving some snow layers a reddish or yellowish color. This is a well-known phenomenon suspected to play an important role on snow covered surface radiative forcing

(De Angelis and Gaudichet, 1991; Di Mauro et al., 2015). Saharan dust depositions, hereafter referred to as dust outbreaks, are very sporadic events mostly occurring from April to August (Varga et al., 2014). They can last only a few hours and drop significant amounts at once creating a strong discontinuity within the vertical profile of snowpack impurity content. This last phenomenon is suspected to affect snow metamorphism and eventually snowpack stability (Landry et al., 2014).

Several snow radiative transfer models accounting for LAI impacts have been developed over the last decades. They provide estimates of spectrally-resolved snow albedo and light penetration in snow for given physical properties of snow and light-absorbing content with various levels of detail. Warren and Wiscombe (1980) and Wiscombe and Warren (1980) established a snow spectral albedo model taking into account the impact of BC and dust. Flanner and Zender (2005) and Flanner et al. (2007) developed another snow spectral radiative model called SNICAR (Snow, Ice, and Aerosol Radiative), based on Wiscombe

and Warren (1980) theory and on the two-stream multi-layer radiative approximation (Toon et al., 1989). The SNICAR model accounts for both the size distribution of LAIs and their location relative to the ice matrix (internal or external mixture). In Carmagnola et al. (2013) and Ginot et al. (2014), DISORT (Discrete Ordinate Radiative Transfer Model Stamnes et al., 1988) was used to compute snow radiative properties in the presence of LAIs both internally or externally mixed. Aoki et al. (2011) developed the Physically Based Snow Albedo Model (PBSAM) which computes the spectral albedo and solar heating

profile within a multi-layer snowpack. In all the previously introduced radiative transfer models, radiative properties of snow corresponded to spherical ice particles. Kokhanovsky and Zege (2004) pointed out that considering snow as spherical particles leads to some errors in the computation of snow radiative properties. They formulated the asymptotic analytical radiative transfer (AART) theory providing analytical formulations for a vertically homogeneous snowpack with non spherical snow particles. This theory has been used in the Two-stream Analytical Radiative TransfEr in Snow model (TARTES Libois et al., 2013) to

compute light penetration and energy absorption in a multi-layer snowpack containing LAIs based on the two stream and $\delta$-Eddington approximations. Malinka (2014) developed a theory to compute spectral radiative properties of a porous material based on the chord length distribution within the snow. This theory was applied to different samples of arctic snow and sea ice snow in Malinka et al. (2016), providing a good estimation of snow spectral albedo in the visible and near infrared wavelength range. Recently, Cook et al. (2017) implemented a radiative transfer model to compute the effect of "red snow algae" on snow

spectral albedo. They used TARTES to compute the spectral albedo of snow containing different types of algae and showed that the impact of algae on snow melt can be greater than that of BC in areas favorable to algae accumulations.

    In order to simulate accurately the radiative properties of an evolving snowpack and to account for the albedo feedbacks, it is necessary to couple radiative transfer models with detailed snowpack evolution models. Coupling intermediate complexity snowpack models accounting for the deposition and fate of LAIs with radiative transfer models was achieved in a few pioneering

studies, which demonstrated that LAI deposition was a major process operating at climate timescales at the global and regional geographical scales. Krinner et al. (2006) showed how dust deposition on seasonal snow cover could impact northern Asia ice cover during the last glacial maximum, using a simple yet pragmatic representation of dust deposition in snow and its impact within the LMDZ4 global climate model. Ménégoz et al. (2014) refined and applied the same land surface model over more recent time periods in order to address the impact of black carbon deposition in snow in the Himalaya region. Flanner

et al. (2007) coupled the snow radiative transfer model SNICAR to a snowpack scheme of the Community Atmosphere Model

global climate model, explicitly simulating BC emissions and transport. This study highlighted the importance of BC in global snow covered surface radiative forcing, showing that the inclusion of BC in snow leads to a global annual mean equilibrium warming up to 0.15°. However, the most detailed snowpack models do not explicitly account for LAI deposition and impact, hitherto. Initial versions of SURFEX/ISBA-Crocus (referred hereafter as Crocus) (Brun et al., 1992; Vionnet et al., 2012) and

SNOWPACK (Lehning et al., 2002) multi-layer detailed snowpack models mostly use empirical albedo decay equations, which do not explicitly account for the deposition of LAI, making them unable to explicitly address LAI/snow physics feedbacks. Jacobi et al. (2015) implemented a radiative transfer scheme simulating dust and BC effects on an Himalayan snowpack simulated with the detailed snowpack model Crocus but in their study the impurity concentration was assumed to be similar in all snow layers and constant over the season. Niwano et al. (2012) implemented a multi-layer snowpack model integrating

PBSAM. This model called Snow Metamorphism and Albedo Process (SMAP) computes radiative properties of an evolving snowpack in which impurities do not evolve; their concentrations are prescribed to field measured values.

Recently Skiles (2014) modified the snowpack model SNOWPACK to track the evolution of dust layers by introducing markers indicating the concentration of dust in each layer. The author implemented a sequential coupling between this snowpack model and SNICAR, run offline. At each time step, the snowpack model computes physical properties needed by SNICAR to

compute the snow broadband albedo offline. This albedo is then re-injected in SNOWPACK at the next time step. This is one of the first attempts to make LAIs evolve inside the snowpack, providing realistic surface LAIs content of the snowpack all along the season. The model they developed computes snowpack evolution under a prescribed dust stratigraphy but does not allow driving the model with atmospheric conditions implying regular LAIs concentrations measurements. Moreover, only the broadband albedo is re-injected in SNOWPACK regardless of the absorption profile which has been proved to have a strong

impact on the temperature profile and in turn on near-surface metamorphism (Libois et al., 2014; Flanner and Zender, 2005; Picard et al., 2016a). Nevertheless, this approach makes it possible to apportion the relative importance of direct and indirect impacts of light-absorbing impurities on energy absorption in snow on a seasonal snowpack. This study shows that in the upper Colorado basin, 80% of LAIs radiative forcing is due to the direct impact against 20% for the indirect impacts, implying that modeling only the snow darkening by LAIs underestimate by 20% their impact.

In order to bridge the gap between detailed snowpack models and LAI deposition, evolution mechanisms and impacts, we implemented LAI deposition and evolution laws in the detailed multi-layer snowpack model Crocus, thereby expanding the reach of such models into assessments of the subtle interplays between snow physics and LAI radiative properties. Taking benefits of the recent inclusion and coupling of the spectral radiative transfer model TARTES (Libois et al., 2015; Charrois et al., 2016) in Crocus, we extended the model capabilities in order to represent LAIs deposition and fate within the snowpack and their direct

and indirect impacts on the snowpack physical properties. In this study, the Crocus model takes typical meteorological driving data required for land surface models measured in the field, complemented by time series of LAI deposition fluxes (BC and dust) extracted from simulations with the ALADIN-Climate atmospheric model (Nabat et al., 2015). Our recent developments on the Crocus model were evaluated for the snow season 2013-2014 at the Col de Porte experimental site (Morin et al., 2012). The results of different simulations with the new developments as well as the original albedo scheme in Crocus are compared

with in-situ field measurements. Finally, the apportionment between direct and indirect impacts of LAIs is estimated. Section 2

details the new developments implemented in Crocus snowpack model and the set-up of the present study. Section 3 introduces the data and methods used to obtain our results and evaluate the model. Finally, the model evaluation and the estimation of the direct and indirect impacts of LAIs are presented in Section 4 and discussed in Section 5.

## 2   Model description

The multi-layer detailed snowpack model Crocus (Brun et al., 1989, 1992) represents the evolution of the snowpack due to its interactions with the atmosphere and the ground. Its input variables are: air temperature, specific humidity and wind speed at a known height above ground; incoming radiation: direct and diffuse short-wave and long-wave; precipitation rate, split between rain and snow. For more details about the snowpack model, a full description of its structure can be found in Vionnet et al. (2012). In the following, we describe the new developments that have been implemented to include LAI-snow interaction
processes which are summarized on Figure 1.

### 2.1   LAI representation in Crocus

Crocus is a Lagrangian model based on numerical snow layers; the snowpack is divided in several layers (up to 50 typically) considered to have homogeneous physical properties (Vionnet et al., 2012). In order to represent the deposition and the evolution of LAIs in Crocus, we created a new prognostic variable corresponding to the mass of LAI present in each layer. For each
Crocus layer, this variable is a one-dimension array representing the mass content of different types of LAI. The model can handle a user-defined number of impurity types characterized by their optical and scavenging properties. In the present study we only focus on two types of LAIs (BC and mineral dust). Deposition and evolution within the snowpack follow several processes, as described below.

#### 2.1.1   LAI deposition

Impurities can be deposited in the snow by two main processes (e.g., Aoki et al., 2006). They can be wet-deposited i.e. atmospheric aerosol particles are scavenged during a precipitation event. Particles present inside or below the clouds are scavenged by hydro-meteors (e.g. rain drops or snow flakes) and deposited on the surface. This deposition mode is represented by scaling LAI content in case of precipitation to the value of the wet deposition flux $W_i$ expressed in g m$^{-2}$ s$^{-1}$. In case of precipitation (solid or liquid), for each type ($i$) of LAI, the mass contained in the precipitation $M_{p,i}$ expressed in g m$^{-2}$ is given
by:

$$M_{p,i} = W_i \times \delta t, \tag{1}$$

where $\delta t$ is the interval time-step of the model in seconds.

In case of snowfall, a new layer of fresh snow is created. The wet-deposited impurity amount is initially assigned to this new layer. In case of rain, the mass of impurity is initially assigned to the uppermost layer.

They can also be dry-deposited by sedimentation or turbulent diffusion, leading to the deposition of aerosol particles on the ground even without precipitation. The parameterization implemented in Crocus considers that the dry deposition affects the near-surface with an exponential decay to take into account wind pumping which buries a fraction of the dry deposited particles by circulating air into the uppermost snow layers. The mass distribution is calculated as follows for each layer ($l$) and each type
($i$) of LAI:

$$M_{t+\delta t,l,i} = M_{t,l,i} + \frac{D_i \times \delta t \times e^{-(z_l/h)}}{\sum\limits_{k=1}^{N} e^{-(z_k/h)} \Delta z_k} \ . \tag{2}$$

Here, $M_{t,l,i}$ and $M_{t+\delta t,l,i}$ represent the mass of impurity type $i$ in g m$^{-2}$ in the layer l at the beginning and end of the time step $\delta t$, $D_i$ the dry deposition flux expressed in g m$^{-2}$ s$^{-1}$ and $h$ is the user-defined e-folding depth characterizing the decrease rate of the impurity distribution with depth. Here $z_l$ is the depth of the layer l and $z_k$ is the depth of the layer k, N being the
total number of Crocus layers. We assume the depth value of a layer to be the distance between the snowpack surface and the middle of this layer. The default value for $h$ is set to 5 mm according to the range of value in Clifton et al. (2008), which shows that wind-pumping affects between 1 and 10 mm of the snowpack surface depending on snow and atmospheric properties. As the typical thickness of the surface layer in Crocus is close to 1 cm, this value of $h$ implies that most of the LAIs are initially deposited in the uppermost layer.

**2.1.2 LAI evolution within the snowpack**

**Handling of layers**

Crocus manages the layers to keep their number under a prescribed maximum value. When there are too many layers, two layers having similar microstructure properties can merge and the properties of the newly created layer are re-calculated (see details in Charrois et al., 2016 or Vionnet et al., 2012). Concerning LAIs content, the impurity mass of the new layer is the sum of the
impurity mass of the two old layers.

On the contrary, when there are fewer layers than the optimum value computed by Crocus, a thick layer ($t$) can be split into two different layers. For each of the newly created layers ($n$), the impurity mass is apportioned according to their snow water equivalent (SWE):

$$M_n = M_t \times \frac{\text{SWE}_n}{\text{SWE}_t}, \tag{3}$$

$M_n$ and $M_t$ being respectively the impurity mass of the newly created and the initial layers in g m$^{-2}$ and SWE$_n$ and SWE$_t$ the SWE of the newly created and the initial thick layer in kg m$^{-2}$.

If a snow layer completely disappears (e.g. due to total melt or sublimation), its impurity content is transferred to the layer below leading to an accumulation of LAI on the top of the snowpack during melt time. This enrichment process has been widely

observed (e.g., Skiles, 2014; Yang et al., 2015). If the disappearing layer is the basal one, its impurity content is discarded by the model.

**Scavenging**

It has been established that some LAI types can be partially scavenged with percolating water during melt time (e.g., Flanner et al., 2007; Doherty et al., 2013; Sterle et al., 2013; Yang et al., 2015). When liquid water percolates into the snowpack, it can carry part of its impurity mass to the layer below. In the current version of Crocus, water percolation is handled following a simple and conceptual bucket approach (Lafaysse et al., 2017). Each layer ($l$) is seen as a homogeneous reservoir containing a given volumetric liquid water content $W_{liq,l}$. For each layer a maximum volumetric liquid water holding capacity $W_{liqmax,l}$ is defined as a percentage of the pore volume. If $W_{liq,l}$ exceeds $W_{liqmax,l}$, the excess water $F_{liq,l}$ drains to the underlying layer.

Similarly to Flanner et al. (2007), we assume LAI inclusion in melt water proportional to its mass mixing ratio multiplied by a scavenging factor. Therefore, a scavenging coefficient $C_{scav,i}$, adjustable for each impurity type ($i$), has been introduced in the model. In case of water percolation, for each layer ($l$) the scavenged mass $M_{scav,i,l}$ is calculated with:

$$M_{scav,i,l} = F_{liq,l} \times C_{scav,i,l} \times \frac{M_{tot,i,l}}{\text{SWE}_l} , \tag{4}$$

where $F_{liq,l}$ is the mass of water leaving the layer $l$ in kg m$^{-2}$ and $\text{M}_{tot,i,l}$ / $\text{SWE}_l$ the impurity mixing ratio, i.e. the ratio between the total mass of impurity of type $i$ in the layer $l$ in kg m$^{-2}$ and the total SWE of the layer in kg m$^{-2}$.

In the present study, we disabled scavenging by default, implying that the default value of BC scavenging coefficient is set to 0%. However in order to assess the impact of BC scavenging we run a configuration implementing a BC scavenging coefficient of 20% according to the values provided in Flanner et al. (2007) and assessed by Doherty et al. (2013) and Yang et al. (2015). Yang et al. (2015) showed that dust particles are too large to be scavenged, consequently mineral dust scavenging coefficient is set to 0%.

## 2.2   Radiative transfer model in snow TARTES

In the original version of Crocus, the albedo is computed for three large spectral bands only and accounting for the properties of the first two snow layers only (Brun et al., 1992; Vionnet et al., 2012). LAIs are not explicitly represented in Crocus original version; their impact is implicitly taken into account by empirically decreasing snow albedo in the visible wavelengths as snow ages.

In this study, the radiative impact of LAIs is explicitly computed using the Two-stream Analytical Radiative TransfEr in Snow (TARTES) (Libois et al., 2013) model, recently implemented in Crocus (Libois et al., 2015). This radiative transfer model computes the spectral absorption of solar radiation within the stratified snowpack using AART theory (Kokhanovsky and Zege, 2004) and the $\delta$-Eddington approximation (Jiménez-Aquino and Varela, 2005). TARTES makes use of four Crocus prognostic variables (SSA, density, snow layer thickness and impurity content) and the angular and spectral characteristics of the incident radiance (solar zenith angle and spectrally resolved diffuse to total irradiance ratio). LAIs are considered to be externally mixed

to the snow and the computation of their radiative impact is based on the Rayleigh approximation (the size of the scattering particles is assumed to be much smaller than the wavelength). TARTES uses the ice-refractive index and two additional variables to characterize each type of LAI : their density and their optical refractive index.

In the present study, we use the value of BC density from Flanner et al. (2012) (1270 kg m$^{-3}$) and the value of mineral dust density from Hess et al. (1998) (2600 kg m$^{-3}$ ). Concerning LAI refractive indexes, values of Chang and Charalampopoulos (1990) are used for BC, as in Libois et al. (2013). Two alternative parameterizations are tested for mineral dust because of the uncertainty of its optical properties. These two parameterizations were taken as an upper and a lower bound on the imaginary part of the refractive index of mineral dust found in the literature. Refractive index values from Müller et al. (2011) are taken as an upper bound of dust absorption and refractive index values from Skiles et al. (2014) are taken as a lower bound of dust absorption. For the ice-refractive index we use the values of Warren and Brandt (2008).

## 2.3 Atmospheric radiative transfer model ATMOTARTES

TARTES requires as input the spectral direct to diffuse incoming irradiance ratio. In SURFEX, it is computed using the newly developed ATMOTARTES scheme, a two stream multi-layer model for atmospheric radiative transfer based on the same two stream code as TARTES (Libois et al., 2013).

The inputs of ATMOTARTES are the atmospheric characteristics : surface pressure and temperature, surface relative humidity, solar zenith angle, day of year, aerosols optical depth at 0.55 $\mu$m, total ozone column (atm-cm), cloud bottom pressure, cloud type (ice or water), cloud optical depth at 0.55 $\mu$m. The cloud optical thickness is diagnosed from the broabdand diffuse and direct solar irradiance estimated from Col de Porte measurements (see Dumont et al. (2017) for more details). The hourly ozone column and aerosols optical depth are provided by ALADIN-Climate. Surface pressure, temperature and relative humidity are provided by the meterological forcings and the solar zenith angle calculation is done within SURFEX. In this study, the scheme is run with 6 layers in the clear sky case and with 7 layers in the cloudy case (the cloud elevation is set to 8 km).

The model is based on three main steps : (i) calculation of the atmospheric optical properties (optical depth, single scattering albedo, and asymmetry factor) for each atmospheric layer, (ii) $\delta$-eddington approximation to account for the forward scattering behaviour of the atmospheric scatterer and (iii) two-stream calculation of the radiative flux. Steps (ii) and (iii) are identical to TARTES. For step (i) parameterization and look-up-tables are taken from Justus and Paris (1985) and Ricchiazzi et al. (1998) to estimate top of atmosphere irradiance, aerosols and clouds optical properties. Rayleigh scattering is computed as in Nicolet (1984) and Bucholtz (1995). Uniformely mixed gas, ozone and water vapour absorptions are computed as in Bird and Riordan (1986). Ozone, water vapour and aerosols vertical profiles are typical mid-latitude winter profiles from SBDART (Santa Barbara DISORT Atmospherice Radiatiave Transfer - Ricchiazzi et al., 1998). SBDART is a plane-parallel radiative transfer model for the atmosphere under clear and cloudy conditions. The solution of the radiative transfer equation is based on DISORT, so is more sophisticated and time consuming than the two flux method used in ATMOTARTES. The model has been evaluated with respect to SBDART (Ricchiazzi et al., 1998) on 1260 different atmospheric profiles. It exhibits a satisfying overall agreement (r$^2$ > 0.988).

## 3 Data and methods

### 3.1 Data and study site

The model simulations and evaluation were carried out at the Col de Porte experimental site for the 2013/2014 snow year. This site is located at 1325 m altitude in the Chartreuse mountain range, France. The model is forced with *in situ* meteorological
measurements from Col de Porte study site namely: air temperature, specific humidity, rainfall and snowfall rates, incident direct and diffuse shortwave radiations, longwave incoming radiation and wind speed. An exhaustive description of the measurement devices and datasets can be found in Morin et al. (2012). Hourly albedo at noon were calculated using spectral reflectance measurements described in Dumont et al. (2017). Measured spectral reflectance were first converted to spectral reflectance for a flat surface using Equation 8 in Dumont et al. (2017). Lastly the spectral reflectance values were integrated over the wavelength
range 350-2800 nanometers, weighted by the incoming spectral irradiance, in order to provide broadband albedo. The same data have been used in Lafaysse et al. (2017) (Figure 1). To constrain LAIs deposition, we use aerosol deposition fluxes from the atmospheric model ALADIN-Climate, a regional climate model based on a bi-spectral semi-implicit semi-Lagrangian scheme (Bubnova et al., 1995). The version 5.3 (Nabat et al., 2015) is used in the present study with a 50 km horizontal resolution, 31 vertical levels and the ERA-Interim reanalysis (Dee et al., 2011) as lateral boundary forcing. For aerosols, no data are available
at the lateral boundaries. Aerosol lateral boundary forcing is set to 0 because ALADIN-Climate domain is considered to be large enough to include all the aerosol sources affecting the area. For instance, the domain includes the whole Saharan desert. ALADIN-Climat includes a prognostic aerosol scheme for the main aerosol species (dust, sea-salt, sulphate, black carbon and organic matter), thus giving an interactive representation of their emission, transport and deposition. Only BC and mineral dust are considered in our snowpack simulations since they are the predominant species in term of radiative impact in the Alps
(Di Mauro et al., 2015). Wet deposition is only activated when there is measured precipitation.

During the 2013-2014 snow year, additional advanced measurements were carried out at Col de Porte. First, chemical analyses of the top of the snowpack were realized on February 11 2014. BC concentration were measured with a Single Particle Soot Photometer (SP2) after nebulization of the meltwater and dust concentrations were measured with a Coulter counter giving vertical profiles from the top 27 cm of the snowpack with 3 cm resolution. Moreover spectral albedos were measured with an
automatic spectroradiometer (Dumont et al., 2017) during the season. The automatic spectroradiometer used was an Autosolexs, whose full description can be found in (Picard et al., 2016b).

### 3.2 Spectral albedo processing

These automatic spectral albedo were processed in order to compute near-surface impurity concentrations and Specific Surface Area (SSA) by Dumont et al. (2017). These data are compared to near-surface properties of snow simulated by the model in the
present study. The model evaluation was performed using the algorithm described in Dumont et al. (2017) applied to Crocus spectral albedo predictions. It accounts for the impact of the top centimeters of the snowpack on spectral albedo, and not only for the Crocus top layer (which is sometimes thinner than the optical e-folding depth). In other words, instead of directly using LAI content and SSA from Crocus top layer, the simulated spectral albedo was used to compute an effective value for near-surface

SSA and equivalent BC content. The equivalent BC content is the concentration of BC in the snow uppermost layers that would have a similar effect on snow spectral albedo that all types of LAIs taken together. Near surface LAI content and SSA are generally not available during snowfall due to large uncertainties in albedo measurements (Dumont et al., 2017).

## 3.3 Model set-up

In this study, all physical options of the Crocus model are set to the default ones as defined in Lafaysse et al. (2017) with the exception of turbulent surface fluxes and surface heat capacity (options RI2 and CV50000). This includes option C13 of the metamorphism scheme implemented by Carmagnola et al. (2014) with prognostic SSA. Hereafter, we refer to the Crocus version using these particular settings as our reference version.

To evaluate the new developments in Crocus we ran different simulations described in Table 1. The configuration C0 corresponds to the reference version of Crocus described above. This configuration does not use the spectral radiative transfer model TARTES but the original parameterization of solar radiation absorption implemented by Brun et al. (1992). The configuration C1 uses the snow radiative transfer model TARTES without impurities while configurations C2, C3, C4 and C5 use TARTES with the new developments. The configuration C2 uses dust refractive index values from Müller et al. (2011) and no scavenging at all. The configuration C3 uses dust refractive index values from Skiles et al. (2014) and no scavenging at all. The configuration C4 uses our new developments with dust refractive index values from Müller et al. (2011) and the scavenging coefficient is set to 20% for BC. Configurations C2, C3 and C4 use BC and dust deposition fluxes from the atmospheric model ALADIN-Climate (more details in Section 3.1).

During the 2013-2014 snow season, two major dust outbreaks occurred in the Alps. Those events are of particular interest for our study as they bring large amount of LAIs at once in the snowpack. First, in mid-February a major dry deposition event struck the Alps. On February 16 a significant wet deposition occurred. Then, on February 19 an intense dry deposition followed leading to a visually observable reddish layer highly concentrated in dust. Secondly, on April 3 another major dry deposition event affected the Alps followed by a significant wet deposition event on April 6.

The configuration C5 uses the same parameterization as C2 but the ALADIN-Climate deposition fluxes were adjusted as follows. For the first dust event, deposition fluxes have been adjusted to match measured dust concentrations at the surface. Indeed, for this outbreak Di Mauro et al. (2015) measured dust concentration ranging from 50 $\mu$g g$^{-1}$ to 330 $\mu$g g$^{-1}$ in the Italian Alps, in a site located approximately 200 km east of Col de Porte at a similar elevation of 1650 m. As dust outbreaks are large scale events, we made the coarse assumption that dust concentration for a same dust outbreak are similar for these two places. For the second major dust outbreak we have not found any measurements so we assumed it has the same magnitude as the first one. We consequently multiplied the dry deposition coefficient by 25 on February 19 and on April 3 for the two major outbreaks and the wet deposition coefficient by 10 on the April 6 to compute a similar deposition. We obtain a deposited near-surface dust concentration of roughly 200 $\mu$g g$^{-1}$ for each event (from 90 $\mu$g g$^{-1}$ to 300 $\mu$g g$^{-1}$ for the first event and from 140 $\mu$g g$^{-1}$ to 350 $\mu$g g$^{-1}$ for the second), consistent with the range of values proposed by Di Mauro et al. (2015). Except for these three days, the deposition fluxes have not been modified. The C5 simulation has been run in order to understand the discrepancies between simulated and measured surface impurity concentrations.

Finally soil temperatures have been initialized by running a single ten-year spin-up, with C0 configuration, from 2003 to 2013 using in-situ meteorological data.

## 3.4 Broadband albedo computation

Lafaysse et al. (2017) hhave shown that Crocus broadband shortwave albedo features a large bias (up to 0.1 depending on the configuration) compared to Col de Porte albedo measurements described in Morin et al. (2012). In order to investigate the origin of this bias we run an additional computation with an offline version of TARTES radiative transfer model. This run uses impurity content simulated with C5 and SSA values retrieved from spectral albedo measurements from Dumont et al. (2017). This simulation is only used in the Section 4.4 and is referred to as "C5(SSA)".

Similarly to the measurements, we only consider broadband albedo computed at noon from downwelling and upwelling broadband radiation fluxes simulated by Crocus. For C0 configuration we use broadband downwelling and upwelling shortwave fluxes at noon to compute the albedo. For the other configurations, we integrate the spectral downwelling and upwelling shortwave fluxes on the shortwave range (300-2800 nanometers) to compute the broadband albedo. Measured and simulated broadband albedo are then compared for days when the simulated snow depth is higher than 0 in all of our simulations and automated spectral albedo measurements are available (46 days in total).

## 3.5 Estimation of direct and indirect impacts

Estimating the portion of LAIs radiative forcing due to the indirect impact requires to separate LAIs evolution and microstructure evolution. With this aim in mind, an additional computation called $C2_{ind}$ was performed, using an off-line version of TARTES. This computation provides snowpack energy absorption using SSA values from C2 simulation while LAI concentrations are set to 0. In this way, energy absorption due to LAI in $C2_{ind}$ only accounts for the accelerated metamorphism disregarding snow darkening (direct impact).

By comparing $C2_{ind}$ computation to C1 (pure snow) and to C2 (full impact of LAIs), we are able to quantify the relative importance of the indirect radiative forcing of LAIs on snow, $R_{ind}$ thanks to the ratio

$$R_{ind} = \frac{E_{C2} - E_{C2,ind}}{E_{C2} - E_{C1}} . \tag{5}$$

$E_X$ being the energy absorbed by the snowpack in configuration X. This ratio can be determined daily $R_{ind,daily}$, or over the whole season $R_{ind,season}$ by applying Equation 5 to the cumulative absorbed energy. Note that the same method can be applied by replacing C2 with C3, C4 or C5.

Our method to compute the LAIs indirect impact is based on the assumption that the total energy absorbed by the snowpack is the sum of the energy absorbed by clean snow and of LAIs impact (direct and indirect). If the ground plays an important role in total energy absorption, our method can not be applied because the influence of the ground may differ between C1 and C2 and cause differences in energy absorption unrelated to LAIs. For this reason all dates with SWE values lower than 50 kg m$^{-2}$ are discarded. This threshold value was obtained by a sensitivity analysis of ground impact on snow visible albedo adapted to

our simulations. For clean snow with high SSA ($> 20$ m$^2$ kg$^{-1}$), it is sufficient to ensure that ground impact is lower than 2 % but for clean snow with low SSA (5 m$^2$ kg$^{-1}$) it would be insufficient (reduction of visible albedo up to 6 %). However in our simulation, at the end of the season the surface snow contains at least 100 ng g$^{-1}$ of BC equivalent, reducing the optical e-folding depth enough to guarantee that the ground does not influence the total energy absorption more than 2 % if the SWE is
higher than 50 kg m$^{-2}$ (even with SSA of 5 m$^2$ kg$^{-1}$).

## 4   Results

### 4.1   Impact of scavenging on the simulated BC vertical profiles

Figure 2 shows the evolution of BC concentration for simulations C2 and C4 during the second half of the season. The differences between these two simulations are only due to the value of BC scavenging coefficient, set to 0% for C2 and 20% for C4. The
BC concentration is almost identical in both cases at the beginning of the period considered, when melt does not occur yet. Then, when melting starts, scavenging decreases BC surface concentration and transfers a part of the BC content to the soil at the bottom of the snowpack (Figure 2b). We can also observe that scavenging transfers a mass of BC from the bottom of the snowpack to the ground all along the season due to basal melt.

### 4.2   Bulk snowpack variables

Figure 3 shows snow depth (upper panel) and snow water equivalent (SWE; lower panel) measured and simulated in the different configurations. Both automatic and manual measurements are shown (represented in black) to illustrate the spatial variability of these variables within the measurement field area because they are not collected at the exact same place.

Snow depth is underestimated by roughly 20 cm compared to automatic measurements for all configurations at the beginning
of the season, from the first snowfall to December 24. Once this initial snowpack has melted, there is a better agreement between observed and simulated snow depth values with all the configurations. The second column of Table 2 presents the RMSE between each simulation and the automatically measured snow depth time series. Over the whole season, the maximum RMSE is 10.0 cm (C1). The third column of Table 2 also presents the RMSE from December 26 to the melt-out date of the snowpack, to better quantify the impact of the configuration on total snow depth estimates disregarding the bias at the beginning of the season.
Over this period, the maximum RMSE is 8.0 cm (C1). It is to note that C1 has also the smallest bias because the underestimation of snow depth during the season (similar to all the other configurations) is compensated by a large overestimation of snow depth from May 20 onward. The values of snow depth bias and RMSE in the present study are consistent with the range of value found for an 18-year period with the recent model uncertainty analysis described in Lafaysse et al. (2017). This value has the same magnitude as the uncertainty of the reference snowdepth as quantified in Lafaysse et al. (2017), as a consequence of spatial
variability. Lafaysse et al. (2017) showed that the automatic snow depth measurements tend to be lower by 9 cm compared to the average of manual snow depth measurements at Col de Porte.

We can also notice that the melt-out date of the snowpack advances by 6 to 9 days when accounting for radiative impact of impurities in snow (comparing C2, C3 and C4 with C1).

Regarding SWE, there is an underestimation in the model during all the snow season compared to both manual and automatic measurements. SWE estimates over the season are similar for all configurations until melt time, when LAIs modify the melting rate. The RMSE between measured and simulated SWE is 90.2 kg m$^{-2}$ for C0 and around 80.0 kg m$^{-2}$ for the other configurations. The minimum RMSE (71.6 kg m$^{-2}$) and bias (64.2 kg m$^{-2}$) are obtained for C1 configuration. There is a significant bias (around 70 kg m$^{-2}$), higher than the magnitude of the reference SWE uncertainty quantified by Lafaysse et al. (2017). However, during this specific season, the automatic snow depth measurements indicates 0 cm of snow on December 26 whereas the SWE automatic measurements indicates more than 70 kg m$^{-2}$ (Figure 3). These results are consistent with Lafaysse et al. (2017) study, which pointed out that spatial variability within Col de Porte site can strongly affect the results of the measurements (about 10 %). It shows that automatic SWE measurements at Col de Porte tend to be higher by 15 kg m$^{-2}$ compared to the average of manual SWE measurements. This process can at least partially explain the relatively low bias obtained for snow depth and the large bias in SWE. A season-specific bias of bulk density is also possible although no long-term bias of this variable was identified by Lafaysse et al. (2017).

## 4.3 Near-surface properties

Figure 4 shows the near-surface impurity concentrations (upper panel) and SSA (lower panel) computed from measured and simulated spectral albedo from February 15 to snow melt-out (around mid-April for all the configurations) by the method described in Section 3.2. These values are computed from processed spectral albedo, C0 (without spectrally resolved albedo) is consequently excluded from the analysis.

The simulated surface impurity content remains within the uncertainties of the indirectly measured data (error bars in the upper panel of Figure 4) except at the very end of the season from April 5 approximately. After this date, the impurity content is lower in Crocus than in the observations. The upper panel of Figure 4 offers an insight into the impact of the parameters modified in the different configurations. The difference between configuration C2 and C4 becomes significant at the very end of the season when strong melting occurs. Before melt time, scavenging does not affect near-surface impurity concentration (Figure 2): C2 and C4 runs give similar results. The difference between C2 and C3 simulations is caused by the different absorption parameterization used for mineral dust. In C3 configuration, dust absorbs less than in C2. The equivalent BC concentration needed to reproduce an equivalent impact on snow albedo is thus lower for C3 when dust is present. In turn, the dates for which C2 and C3 are similar correspond to situations when mineral dust is not the dominant absorber.

The Crocus near-surface SSA decreases too slowly after a snowfall under Col de Porte meteorological conditions, regardless of the configuration (Figure 4, lower panel). The decrease rate of SSA is computed using the C13 metamorphism scheme implemented by Carmagnola et al. (2014), untouched in this study. However, it is clear that the impacts of LAI modifies the SSA

decrease rate. Indeed with C1 configuration the bias between measured and simulated near-surface SSA is -4.9 m$^2$ kg$^{-1}$ against -4.2 m$^2$ kg$^{-1}$ for the configurations implementing LAIs. Figure 4 highlights that SSA values for C2, C3 and C4 are almost the same, indicating that the different LAI parameterizations used in this study have a negligible impact on surface SSA evolution.

## 4.4 Broadband shortwave albedo

Figure 5 shows the evolution of the simulated and measured broadband albedo at noon. The simulated broadband albedo is higher than the measurements for all the configurations except for C5(SSA) for which SSA values have been adjusted to measured ones.

The last column of Table 2 provides albedo bias and RMSE resulting from this comparison. Those results are consistent with RMSE/bias values obtained in Lafaysse 2017 ensemble simulation. Except for C5(SSA), C0 outperforms the other configurations
in terms of albedo. Equivalent scores are obtained for C5 configuration and the difference between C1 and C2, C3, C4 shows that accounting for LAI largely improve the albedo simulations over a simulation neglecting the impact of impurities. Albedo bias for C5 simulation is significantly reduced by using measured SSA values instead of the simulated ones, suggesting that the albedo bias is partly explained by the bias in SSA.

## 4.5 Profiles of impurity concentration

Figure 6 shows vertical profiles of BC and dust content in the top 25 centimeters of the snowpack on February 11 both measured and simulated with configurations C2 to C4. BC concentrations are significantly overestimated and dust content are underestimated. Moreover, the vertical structure is not correctly reproduced. It is to note that in our simulation, the uppermost 17 cm of snow correspond to a unique snowfall that occurred on February 10. During this snowfall ALADIN-Climate did not simulate any mineral dust deposition explaining the absence of dust in this part of the snowpack.

## 4.6 Quantification of direct and indirect LAI radiative impact

The upper panel of Figure 7 shows the energy absorbed by the snowpack for C1, C2 and C2$_{ind}$; the grey shaded areas indicates period during which SWE is less than 50 kg m$^{-1}$ in both simulations. The daily energy absorption (full lines) shows that the total radiative impact of LAIs increases through the season (difference between the red and the green curves). Applying Equation 5 on the cumulative energy absorbed during the season (dashed lines) provides the ratio of LAI radiative impact due to the
indirect impact over the whole season. R$_{ind,season}$. Indeed, applying Equation 5 on the total cumulative energy absorbed at the end of the study period, we determine that over the season, 15.3% of LAI radiative forcing is due to the indirect impact (R$_{ind,season}$), while 84.7% of LAI impact is caused by the direct impact. The lower panel in Figure 7 shows the daily percentage of LAIs radiative forcing caused by the indirect impact along the snow season. The values potentially affected by the ground (in orange) have to be taken with caution because the ground influence might have modified the results. These results are shown in
parallel to the value of the SSA because the indirect impact of LAIs is due to an acceleration of snow metamorphism meaning an acceleration in SSA decrease rate.

Sections 4.2 and 4.4 highlight that C5 provides better results than C2 in terms of near-surface LAIs concentration and shortwave albedo. Given that radiative forcing is expected to be more accurate for C5, the same method has also been applied using C5 as a control run (instead of C2 in Figure 7). We obtain similar results in term of temporal evolution but the distribution between the average direct and indirect impacts is only slightly modified, with 14.1% attributed to the indirect impact instead of 15.3%, which we consider an insignificant variation..

## 5    Discussion

### 5.1    Simulated LAI contents

Section 4.5 highlighted discrepancies between simulated and measured dust and BC vertical profiles for February 11 2014. BC content simulated by the model is an order of magnitude higher than the measured BC content. In contrast the dust content simulated by the model is an order of magnitude lower than the measured dust content. For both types of LAIs the vertical structure is not reproduced. Several hypotheses can explain these discrepancies.

First, ALADIN-Climate has a 50 km horizontal resolution which cannot represent the local orography around Col de Porte site. Hence, the atmospheric variables in the model (e.g. wind, precipitation rate) do not account for small scale topography which is particularly important in mountain areas. For example, in ALADIN-Climate local dust erosion is represented as a function of wind and soil characteristics. If the wind on the grid point is low but small scale phenomena induce stronger winds near the Col de Porte, the resulting soil erosion and transport are not caught by ALADIN-Climate. This last point can explain partly or totally the strong underestimation of mineral dust concentration in the model.

Secondly, the Col de Porte experimental site is located near Grenoble, France which is a city affected by high levels of air contamination (Maître et al., 2002). However, Col de Porte is more than 1000 m higher in elevation than Grenoble. The difference between simulated and measured BC concentration vertical profiles may come from an overestimation of Grenoble's impact on Col de Porte study site by ALADIN-Climate. The deposition fluxes extracted from ALADIN-Climate correspond to a grid cell associated with an elevation of 523 m of elevation, an altitude difference of about 800 m. Even if this cell does not include Grenoble, it may explain partially the overestimation of BC deposition by the model. Moreover persistent winter inversions are frequently observed in Grenoble. These phenomena could lead to accumulation of BC emissions in the lower part of the atmosphere, preventing significant transport to Col de Porte. ALADIN Climate can not represent these winter inversions because of their relative small-scale compared to the model resolution. This may also partly explain the overestimation of BC deposition fluxes predicted by the model.

Even though the vertical impurity concentration profiles on February 11, presented above, are not correctly simulated, the near-surface BC equivalent computed from simulations are in good agreement with the one computed from measured spectral albedo except at the end of the season (from April 5). The main cause of the divergence at the end of the season might be an underestimation of the two major Saharan dust outbreaks by ALADIN-Climate. The chronology of major dust outbreaks for snow year 2013-2014 is presented in Section 3.3 (see Figure 8).

A plausible assumption is that the amount of dust deposited by each of these two major dust outbreaks at Col de Porte are underestimated by ALADIN-Climat. The divergence may be due to both the underestimation of the April dust outbreak and the reappearance of the dusty layer formed on February 19 event (around April 8) after ablation of the overlying layers (Figure 8). This assumption could explain why near-surface impurity contents fit the measurements before April 3 and diverge after this date.

The upper panel of Figure 4 points out that C5 improves the simulated late season near-surface impurity concentrations compared to all other configurations. However, in order to test this hypothesis a more detailed evaluation of the LAI (BC and dust) contents in snow should be performed using direct measurements of LAI and not LAI content estimated from (hyper)spectral measurements (e.g., Warren, 2013) which are uncertain for low impurity content (Dumont et al., 2017) but is beyond the scope of the present study.

The divergence on the late-season near-surface LAI concentrations may also come from the impact of the neglected LAI types such as organic debris (which are present at Col de Porte) or brown carbon. Additional chemical analyses would be required to investigate both these assumptions.

Lastly, it must be underlined that the wet deposition fluxes from ALADIN-Climate are only taken into account in the simulations when in-situ precipitation is measured. Consequently, any mismatch between ALADIN-Climate and measured
precipitation occurrence may lead to errors in simulated wet deposited LAI content.

## 5.2    Impact on Crocus melting rate

Through the new developments implemented in Crocus we evaluate the impact of LAIs on the melting rate for the 2013-2014 snow season at Col de Porte. We show that the melt-out date of the snowpack advances by 6 to 9 days when accounting for radiative impact of impurities in snow (Figure 3).

In the reference version of Crocus (C0), LAIs in snow are implicitly taken into account by decreasing the albedo in visible wavelengths as snow ages. This albedo decrease has been implemented to empirically fit the snow melting rate under meteorological conditions observed at Col de Porte, which has been the main evaluation site of Crocus model. This explains why the initial version is in agreement with the observations and the new developments do not imply a direct visible improvement. However, as illustrated by Lafaysse et al. (2017), this albedo parameterization and the calibration of its characteristic time
constant are rather uncertain. This uncertainty is addressed by the physically based parameterization presented in this study which can moreover account for regional and temporal variability of LAI deposition.

When using the TARTES radiative transfer model, the impurities are explicitly taken into account and there is no empirical albedo reduction due to snow aging. This explains why C1 configuration (TARTES without impurities) underestimates the melting rate. The inclusion of impurities improves albedo computation and in turn snow melting rate at the end of the season.
The atmospheric deposition fluxes provided by ALADIN-Climate (C2,C3 and C4) improve melting rate at the end of the season compared to C1 simulation although SWE is simulated more accurately using C1, probably due to a bias at the beginning of the season. The second column of Table 2 presents RMSE on snow depths for the end of the season (January to melt). RMSE is around 6 cm for C0, C2, C3 and C4 showing that similar results are obtained in term of late-season melting rate with both the

new physically based albedo scheme described in this study and the empirically based original scheme. A comparison in other sites are more likely to show discrepancies between the two approaches as the original scheme was calibrated at Col de Porte.

However, even if C5 improves the near-surface impurity concentration, the melting rate increases too much, which worsens the SWE and snow depth simulations. A better simulation of the amount of LAIs in snow thus leads to overestimating the melt rate. This may come from the high equifinality in snowpack modeling as pointed out in the conclusion of Lafaysse et al. (2017). Indeed snowpack models contain several empirical parameterizations, each introducing modeling errors couterbalancing each other and yielding consistent results. For this reason, improving a process in the snow model does not necessarily improve the snowpack simulations.

### 5.3   Direct and indirect radiative impact of LAIs

We estimate that over the whole season 2013-2014, about 15% of the LAIs radiative forcing comes from the indirect impact while 85% is due to the direct impact for the C2 configuration. This means that models which do not represent snow metamorphism and only account for the direct impact of LAIs underestimate by approximately 15% the radiative forcing of LAIs on snowpacks with similar characteristics to Col de Porte. These results are close to the ones presented in Chapter 5 of Skiles (2014) showing that in the Colorado upper basin, 80% of LAI radiative forcing comes from the direct impact against 20% for the indirect impact. The discrepancy is small and might be explained by the differences between the two studies (e.g. different LAI type, different atmospheric conditions, different snow SSA and different unfolding of the season) as the relative contributions of direct and indirect impacts have a period and site dependency.

When looking at the lower panel in Figure 7 we can notice some patterns in the evolution of the percentage of indirect impact according to SSA. Indeed, after a snowfall (resulting in high surface SSA), the SSA decreases quickly due to accelerated snow metamorphism. In this period of fast metamorphism the indirect impact is particularly high (up to 60% on March 7) because the small additional energy income due to LAIs in fresh snow leads to an accelerated SSA decrease. After reaching a value around 10 m$^2$ kg$^{-1}$, SSA decreases much slower and the indirect impact becomes small (below 10%). Then, from March 13 the snowpack is affected by a period of intensive melt, leading to low SSA (around 8 m$^2$ kg$^{-1}$). This SSA decrease is amplified by LAI radiative forcing as surface LAI content at the surface is relatively high during this period (Figure 4), leading to even lower SSA. This additional SSA decrease caused by LAIs cause an increased indirect LAI radiative forcing (up to 25% on March 20). We can then observe the same pattern around April 15.

### 5.4   Shortwave albedo computation

Section 4.4 highlights that shortwave albedo computation features a significant bias for all the configurations, also noticed by Lafaysse et al. (2017) regardless of the albedo scheme employed. Snow albedo is not only dependent on snow LAI contents but also largely depend on SSA values, which have been shown to exhibit a 4 m$^2$ kg$^{-1}$ bias for near-surface snow. The additional computation run using optimized SSA values indicate that most of the albedo bias is due to the bias in SSA (last column of table 2). Modifications of other Crocus parameterizations (such as the SSA evolution laws) would therefore be required to significantly improve shortwave albedo computations.

Section 4.4 also points out that our recent developments do not improve the albedo computation compared to the reference version (C0 compared to C2, C3, C4 and C5). However, these developments are expected to improve Crocus shortwave albedo computations if they were applied to regions with different contamination levels of LAIs compared to the Col de Porte (e.g: Colorado, Arctic, Antarctic... ) where the reference empirical albedo scheme calibrated at Col de Porte poorly performs.

Finally, as underlined in Lafaysse et al. (2017) the improvement of one parameterization does not necessarily lead to the improvement of the overall snow simulations. For example, snowdepth evolution at Col de Porte is simulated reasonably despite a strong shortwave albedo overestimation. This albedo bias is compensated by other parameterization biases; correcting this bias would hence lead to a degradation of snowpack simulation if the other parameterizations stay untouched (e.g C5 compared to C2, C3 and C4).

## 5.5 Model limitations

The parameterization of liquid water content in Crocus follows a simple conceptual bucket approach which does not represent accurately the evolution of liquid water content in the snowpack, as pointed out in Lafaysse et al. (2017). Work is in progress to include a physically based liquid water parameterization in Crocus (D'Amboise et al., 2017). Changing the liquid water content parameterization is expected to improve the modeling of water percolation and impact the scavenging of LAIs in the snowpack. Indeed physically based approaches induce much more heterogeneous repartition of the liquid water content at melt time than the bucket approach (e.g. due to the representation of capillarity barriers ; Wever et al., 2014 ) . We would therefore expect a more realistic and heterogeneous LAIs repartition after scavenging.

Concerning atmospheric radiative transfer (Section 2.3), ATMOTARTES only has a rough representation of the effect of LAIs in the atmosphere (one type of aerosols and constant vertical profile). This could be extended as in SBDART (Ricchiazzi et al., 1998) but the impact would be limited while the numerical cost would be significantly increased.

Several model and parameter choices relative to in-snow radiative transfer also contain some limitations. First, here we use the ice refractive index value proposed in Warren and Brandt (2008) but alternative parmeterizations could also be used (e.g. the visible range parameterization proposed in the recent study of Picard et al., 2016a) and impact the results. Secondly, LAIs are represented as Rayleigh scatterers in TARTES (their size is assumed much smaller than the wavelength). This theory is acceptable in the case of BC but may not perfectly apply to dust, depending on its volume size distribution, and may lead to an underestimation of dust radiative impacts. Coulter measurements show that the average diameter according to their volume contribution for our dust is 2.8 μm, which indeed suggest that dust radiative impact is underestimated here and calls for another parameterization of LAI impacts in TARTES for dust particles. Finally, in the present study LAIs are assumed to be externally mixed to the ice matrix. Flanner et al. (2012) showed that internally mixed BC was up to 80% more absorptive than externally mixed BC. Recently, Liou et al. (2014) and He et al. (2014) also pointed out that both impurity-snow internal mixing and snow nonsphericity play very important roles in snow albedo calculations. They showed that internal mixing can enhances BC-induced snow albedo reduction up to 50% compared with external mixing. This enhancement is stronger for nonspherical ice elements than ice spheres, although ice spheres still have a larger absolute albedo reduction than nonspherical ice elements under the same BC content in snow. Introducing an internally-mixed representation of LAIs in TARTES could in turn impact the results.

However, a better knowledge of the partition between internally and externally mixed LAIs in seasonal snowpacks would be required to accurately characterize the impact of this variable.

## 6    Conclusion and outlooks

In this study, new developments aiming at modeling the deposition and the evolution of light absorbing impurities (LAIs) within the snowpack are introduced in the detailed snowpack model Crocus. We implemented the dry and wet deposition of a user-defined number of LAI species. The deposition fluxes can either be extracted from an atmospheric model as in this study, or forced by user prescribed deposition rates as in Charrois et al. (2016). The fate of the aerosols deposited in the snow is computed by mass-conservation evolution laws for impurity mass content as snowpack evolves. Finally, we use the radiative transfer model TARTES embedded into Crocus to explicitly account for the direct and indirect radiative impact of the LAIs evolving in the snowpack.

This newly implemented Crocus version was then evaluated with field measurements performed at the Col de Porte experimental site (French Alps) near Grenoble, during the 2013-2014 snow year. For this evaluation we accounted for two LAI species assumed to have the strongest radiative impact on snow: BC and mineral dust. We extracted aerosol deposition fluxes from the atmospheric model ALADIN-Climate and forced the snowpack model with these deposition values. We evaluate the relevance of using atmospheric aerosol with a physically based model in terms of near-surface impurity concentration, near-surface SSA, snow depth and SWE. It appears that the atmospheric model ALADIN-Climate as a forcing data-set simulates LAI deposition acceptably over a season despite a large under-estimation of extreme dust outbreaks and an overestimation of BC deposition. Radiative transfer properties of a seasonal snowpack in the presence of dust and BC can be computed efficiently following a physically based approach coupled to atmospheric aerosol deposition fluxes.

The impact of LAIs in term of snow height and SWE is significant. Indeed, depending on the configuration chosen for LAI parameters, complete snow melt out date advances by 6 to 9 days in comparison with the pure snow simulation. This impact on snow melting is of crucial importance for hydrological concerns. We also estimate the direct/indirect proportion of LAI radiative forcing. For Col de Porte in this particular season 85% of the radiative forcing of LAIs in snow comes from the direct impact (darkening of the snow) against 15% for the indirect impact (enhanced metamorphism). This means that models representing LAIs radiative impact of snow without accounting for the metamorphism underestimate by 15% of the total impact. Moreover at daily resolution, the relative proportion of direct and indirect impacts can vary widely, showing evolution patterns in link with SSA evolution.

Our study highlights the need for intensive field campaigns to better evaluate these new developments and better understand these processes. Some parameters of our newly implemented version still need to be adjusted towards field data currently missing. Concomitant measurements of snow temperature, SSA, accumulation of soot and dust, and spectral albedo at different sites would provide a stronger basis for defining model parameters and evaluating it. For example, a direct evaluation of the dust and BC contents is required to quantify more precisely their respective part in the shortening of the snow season.

We showed that the use of atmospheric aerosol deposition fluxes provided by ALADIN-Climate coupled with the recent developments of Crocus leads to a reasonable estimation of snow surface impurity content. Even if this estimation is not perfect due to modeling uncertainties and atmospheric model horizontal resolution, it gives a first guess of LAI impacts on snow spectral albedo. This first guess is a crucial point for assimilating optical reflectance measurements in a snowpack model although a better quantification of the errors in the impurity forcing and modelling will be required (Charrois et al., 2016).

This study is one of the first attempts to account for the deposition and the evolution of impurities in a detailed snowpack model. Here we investigate the effect of dust and BC on snow radiative properties at the Col de Porte experimental site but our model can apply to any snow-covered regions affected by LAIs. This model could be used in dust-affected areas (e.g. Colorado or Himalaya) or BC-affected regions (e.g. Artic or Antarctic regions for climate studies). It could also be use to assess the impact of ashes on snow in volcanic regions (e.g. Iceland). Moreover, Crocus provides habitat data for in-snow ecological modeling (e.g. snow temperature, liquid water content). With the recent developments presented in this study it could be envisaged to compute nutrient evolution in the snowpack. Then, it appears possible to model algae growth, evolution and radiative impacts (Cook et al., 2017) on the snowpack.

Finally, Crocus is now capable of tracking thin layers highly concentrated in LAIs (e.g. Sarahan dust outbreaks) in the snowpack and representing the discontinuity induced in terms of energy absorption and thus snow metamorphism. Our new developments could then be used to address numerically the frequently asked question: "Is there a link between dust outbreaks and avalanche hazard?" (Landry, 2014, Chomette et al., 2016).

## Author contributions

M. Dumont and F. Tuzet coordinated the study. F. Tuzet, M. Dumont, L. Charrois, M. Lafaysse and S.Morin implemented and tested the new developments in Crocus snowpack model. M. Dumont developed the near-surface properties computation algorithm. G. Picard and L. Arnaud developed and built the automatic albedometer. G. Picard, L. Arnaud, S. Morin and D. Voisin deployed it at Col de Porte. D. Voisin performed the impurity content measurements. P. Nabat provided the ALADIN-Climate simulations and Y. Lejeune provided the snow and meteorological measurements at Col de Porte. F. Tuzet prepared the manuscript with contributions and feedbacks from the other authors.

## Acknowledgments

CNRM/CEN and IGE are part of Labex OSUG@2020 (investissement d'avenir - ANR10 LABX56). This study was supported by the ANR programs 1-JS56-005-01 MONISNOW and ANR-16-CE01-0006 EBONI; the INSR/LEFE projects BON and ASSURANCE; the Ecole Doctorale SDU2E of Toulouse. The authors are grateful to the Col de Porte and EDF/DTG staff for ensuring a proper working of all the instruments, to L. Mbemba for the in situ measurements of impurity content.

**Code availability**

The code used in this study is developed inside the opensource SURFEX project (http://www.umr-cnrm.fr/surfex). While it is not implemented in an official SURFEX release, the code can be downloaded from the specific branch of the svn repository maintained by Centre d'Études de la Neige. The full procedure and documentation can be found at https://opensource.

5 cnrm-game-meteo.fr/projects/snowtools/wiki/Procedure_for_new_users. For reproductibility of results, the version used in this work is tagged as https://opensource.umr-cnrm.fr/projects/surfex_git2/repository?utf8=%E2%9C%93&rev=tuzetTCD17.

**Data availability**

The Col De Porte dataset is placed on the PANGAEA repository (doi 10.1594/PANGAEA.774249) as well as on the public ftp server ftp://ftp-cnrm.meteo.fr/pub-cencdp. Time series of snow spectral albedo and superficial snow-specific surface area and

10 impurity content are available through the PANGAEA database (doi:10.1594/PANGAEA.874272).

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

**Tables**

| | Settings | BC | | Dust | |
|---|---|---|---|---|---|
| | | Scavenging | Optical properties | Scavenging | Optical properties |
| C0 | Reference version | / | / | / | / |
| C1 | TARTES without impurities | / | / | / | / |
| C2 | TARTES with ALADIN-Climate deposition fluxes | 0% | Chang and Charalampopoulos (1990) | 0% | Müller et al. (2011) |
| C3 | TARTES with ALADIN-Climate deposition fluxes | 0% | Chang and Charalampopoulos (1990) | 0% | Skiles et al. (2014) |
| C4 | TARTES with ALADIN-Climate deposition fluxes | 20% | Chang and Charalampopoulos (1990) | 0% | Müller et al. (2011) |
| C5 | TARTES with ALADIN-Climat modified deposition fluxes (accounting for dust outbreaks) | 0% | Chang and Charalampopoulos (1990) | 0% | Müller et al. (2011) |

**Table 1.** Crocus configurations used.

| Configuration | Depth | | SWE | Near-surface SSA | Broadband shortwave albedo at noon |
|---|---|---|---|---|---|
| | RMSE(bias) from 05/11/13 to 01/05/14 | RMSE(bias) from 26/12/13 to 01/05/14 | RMSE(bias) from 05/11/13 to 01/05/14 | RMSE(bias) from 15/02/13 to 15/04/14 | RMSE(bias) from 15/02/13 to 15/04/14 |
| C0 | 8.5(-6.9) cm | 6.4(-5.3) cm | 90.2(-79.1) kg m$^{-2}$ | X | 0.059(+0.049) |
| C1 | 10.0(-2.7) cm | 8.0(+1.2) cm | 71.6(-64.2) kg m$^{-2}$ | 7.6(+4.9) m$^2$ kg$^{-1}$ | 0.121(+0.094) |
| C2 | 8.9(-6.1) cm | 6.0(-3.8) cm | 84.4(-75.0) kg m$^{-2}$ | 6.9(+4.2) m$^2$ kg$^{-1}$ | 0.078(+0.060) |
| C3 | 8.8(-5.9) cm | 5.8(-3.4) cm | 82.9(-74.0) kg m$^{-2}$ | 6.9(+4.1) m$^2$ kg$^{-1}$ | 0.078(+0.061) |
| C4 | 8.8(-5.9) cm | 5.9(-3.5) cm | 83.4(-74.3) kg m$^{-2}$ | 6.9(+4.2) m$^2$ kg$^{-1}$ | 0.081(+0.063) |
| C5 | 9.0(-6.4) cm | 6.2(-4.1) cm | 85.6(-75.8) kg m$^{-2}$ | 6.9(+4.3) m$^2$ kg$^{-1}$ | 0.067(+0.054) |
| C5(SSA) | X | X | X | X | 0.044(+0.020) |

**Table 2.** RMSE and bias between measured and simulated variables. For snow depth and SWE, the RMSE and bias are computed from the automatic measurements. The SSA values are computed from the spectral albedo both measured and simulated. The spectral albedo computation is not activated in the reference Crocus version (C0), explaining the lack of RMSE and bias values for the corresponding box.

**Figures**

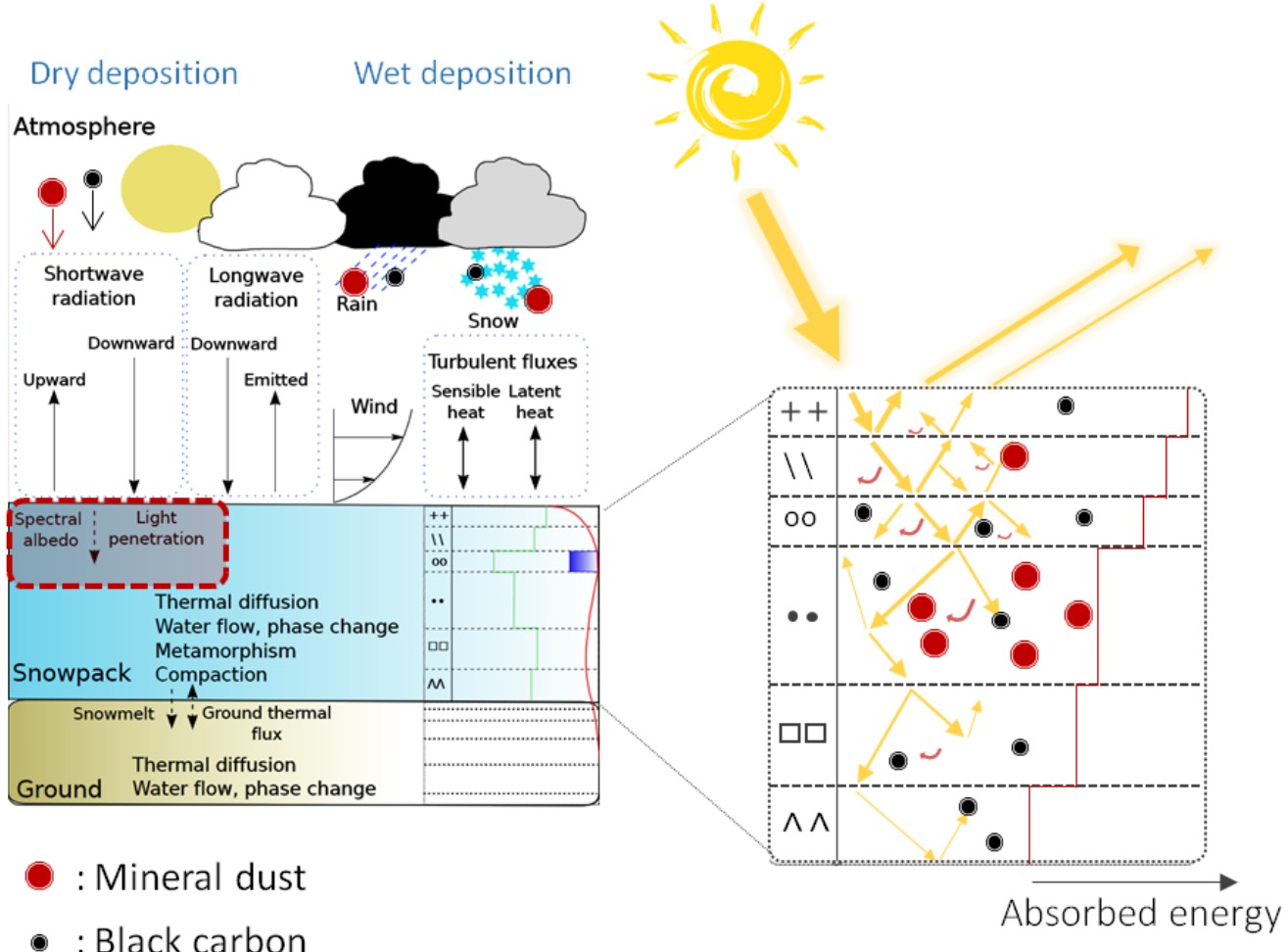

**Figure 1.** Description of the detailed snowpack model Crocus including an explicit representation of LAIs deposition and evolution.

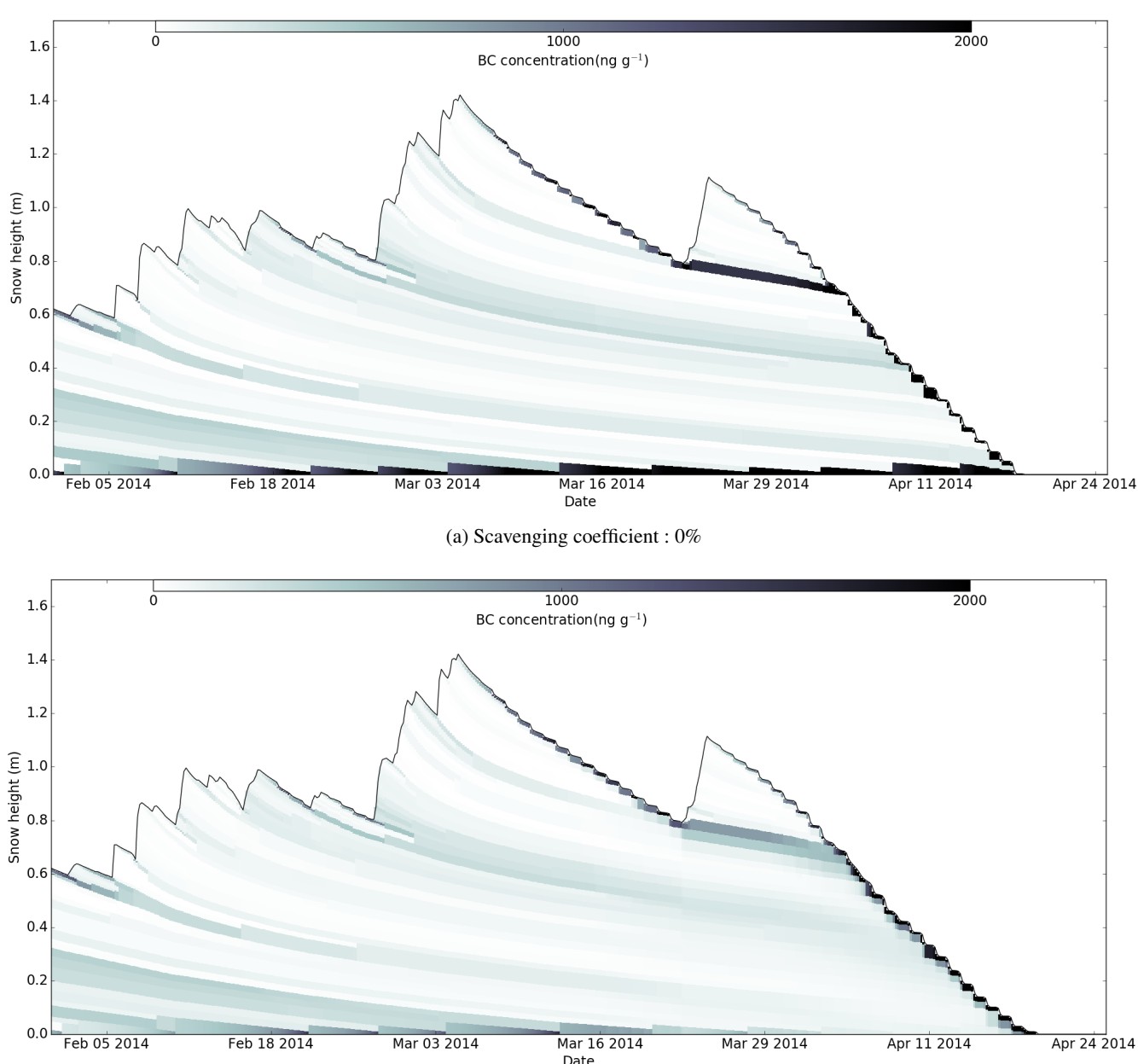

(a) Scavenging coefficient : 0%

(b) Scavenging coefficient : 20%

**Figure 2.** Simulated BC concentration evolution at the end of 2013/2014 snow season at Col de Porte. The upper panel corresponds to a simulation without scavenging whereas the lower panel corresponds to a simulation using the value of 20% for BC scavenging.

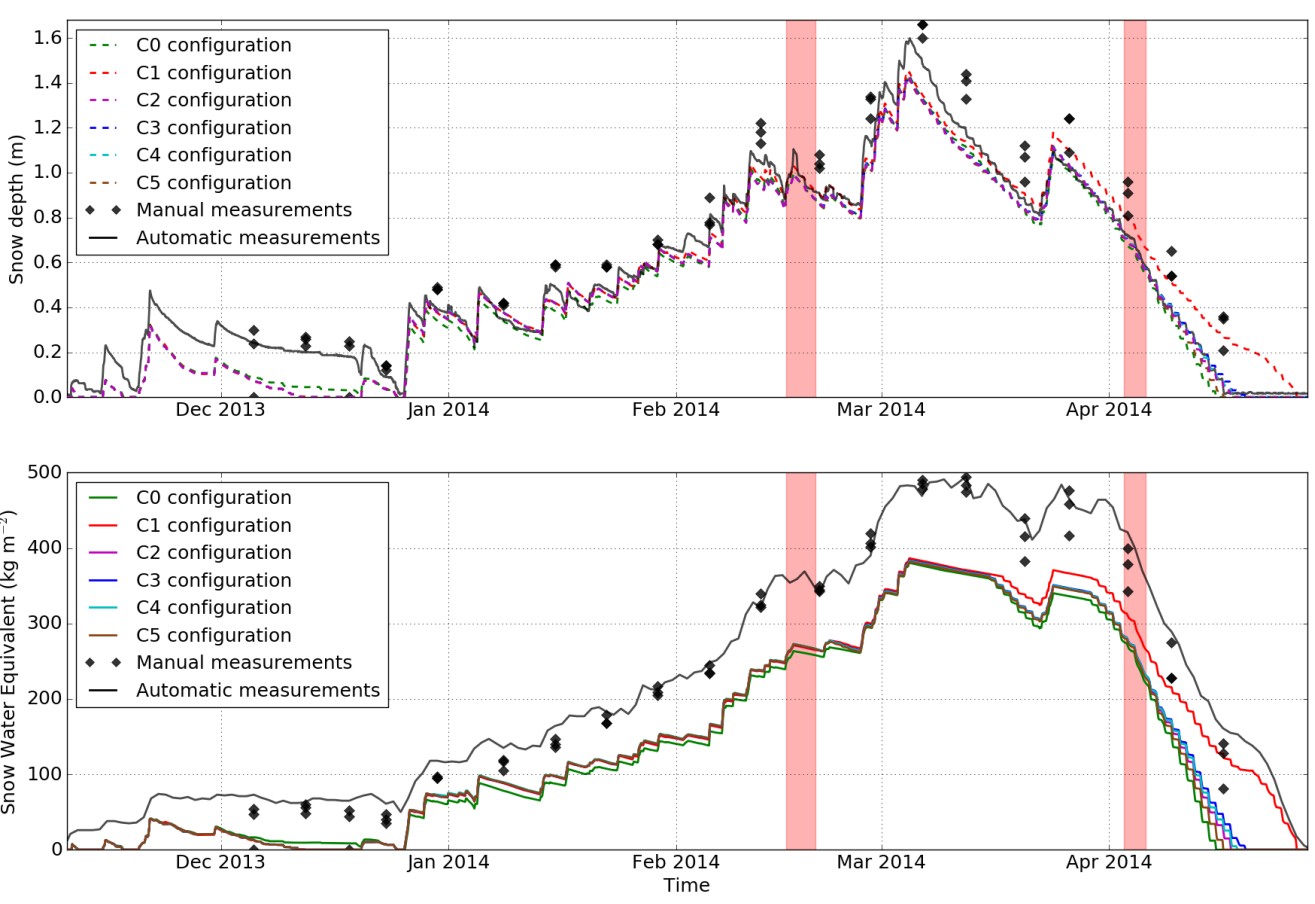

**Figure 3.** Measured and simulated total snow depth (upper panel) and total SWE (lower panel) at Col de Porte along 2013-2014 snow year. The two major Saharan dust events are represented by the red shading.

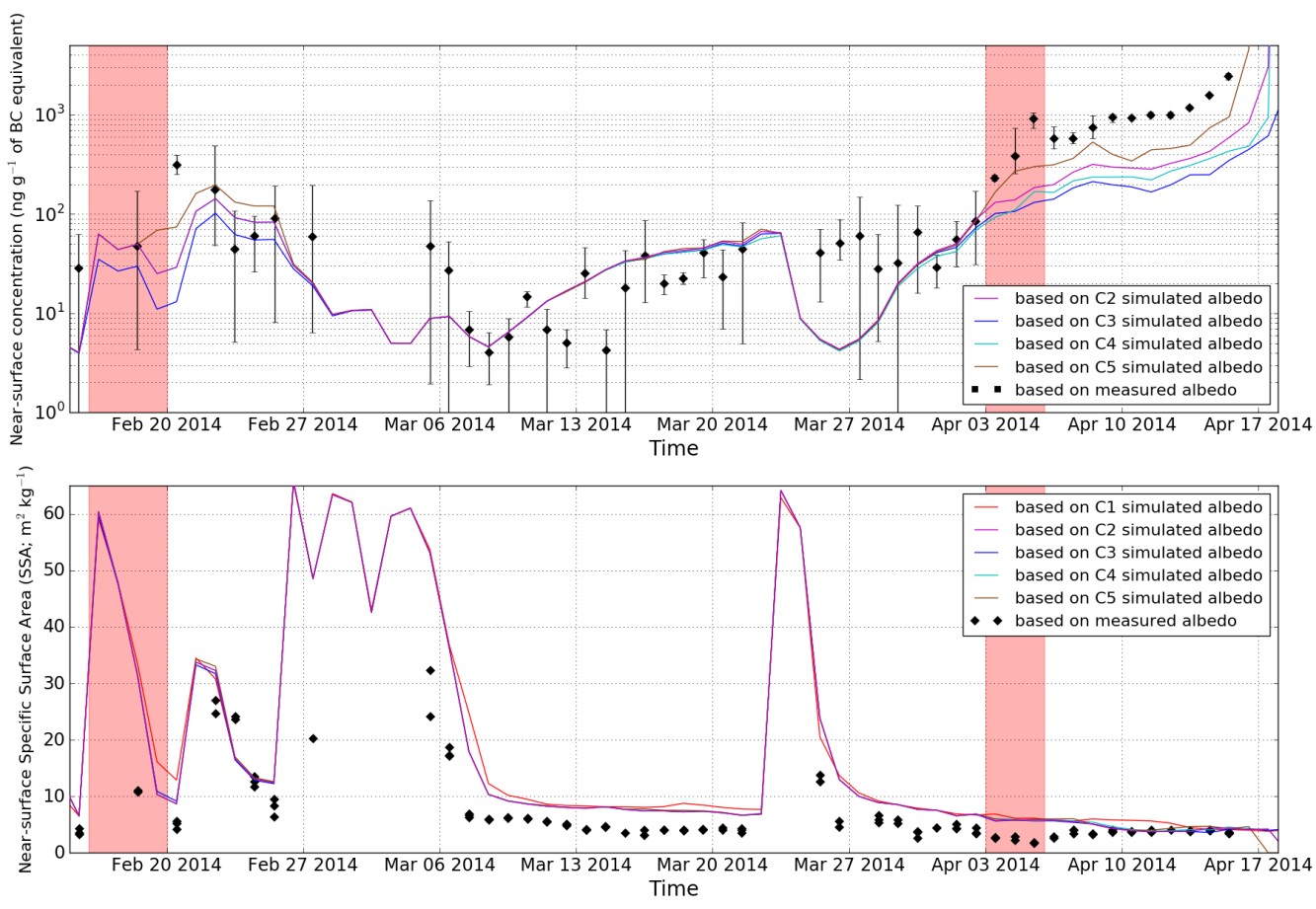

**Figure 4.** Surface BC equivalent concentration (uper panel) and SSA (lower panel) computed from mesured and simulated albedo. For simulated albedo, the different Crocus configurations are detailed in Table 1. These data have been computed using Dumont et al. (2017) algorithm and the two major Saharan dust events are represented by the red areas.

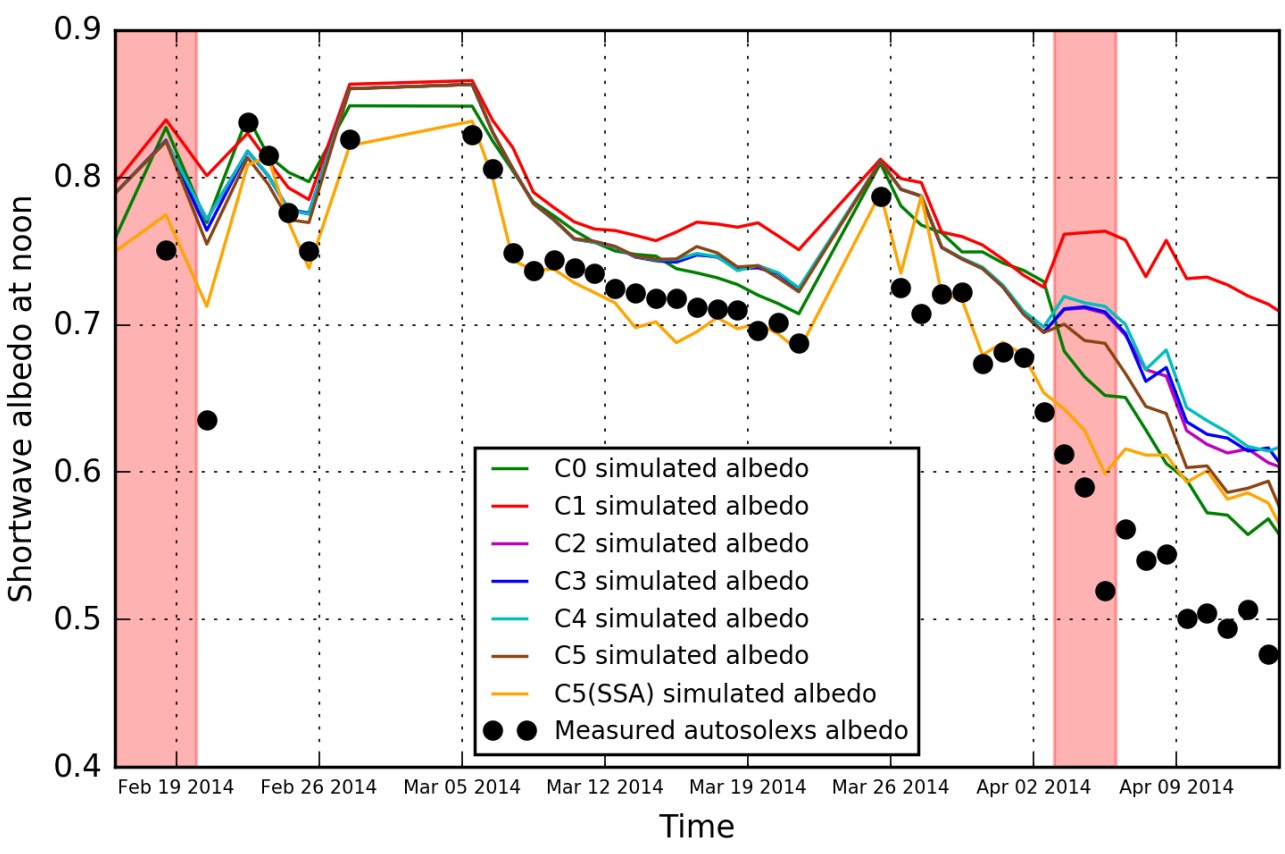

**Figure 5.** Shortwave broadband albedo at noon. The colored lines correspond to simulated albedo while the black dots correspond to Autosolexs measured albedo (Dumont et al., 2017). The two major Saharan dust events are represented by the red shading.

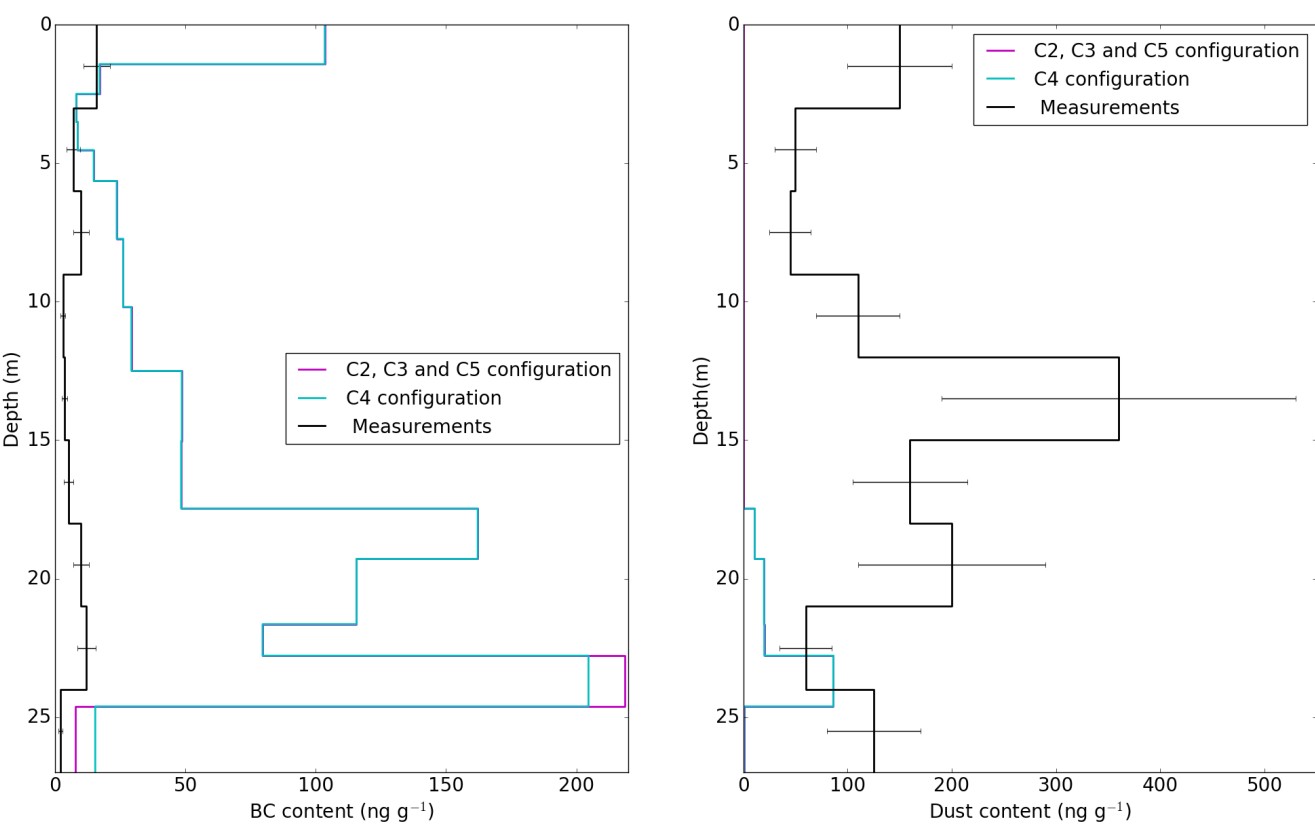

**Figure 6.** BC (left) and dust (right) concentrations at Col de Porte on the 11 February 2014.

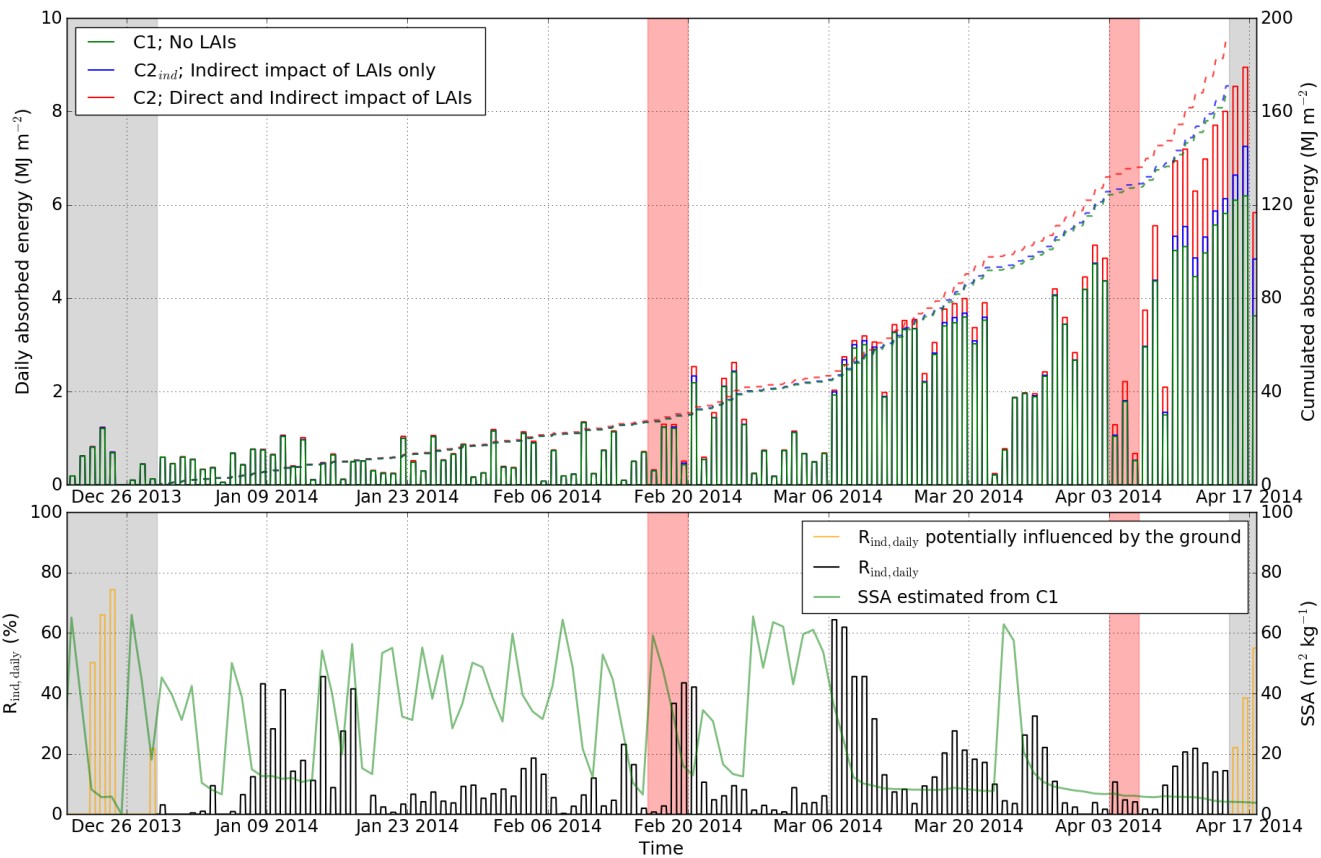

**Figure 7.** Energy absorbed by the snowpack during the season (upper panel); the full lines correspond to the daily amount of energy absorbed whereas the dashed lines corresponds to the cumulative energy absorbed over the study period. $R_{ind,daily}$ compared to near-surface SSA computed from C1 (lower panel); $R_{ind,daily}$ is the daily relative importance of LAIs in snow radiative forcing coming from the indirect impact (Equation 5 applied to daily enrgy absorption). The dates during which the ground influences the energy budget have been masked (grey shading). The red shading represents two major Saharan dust events.

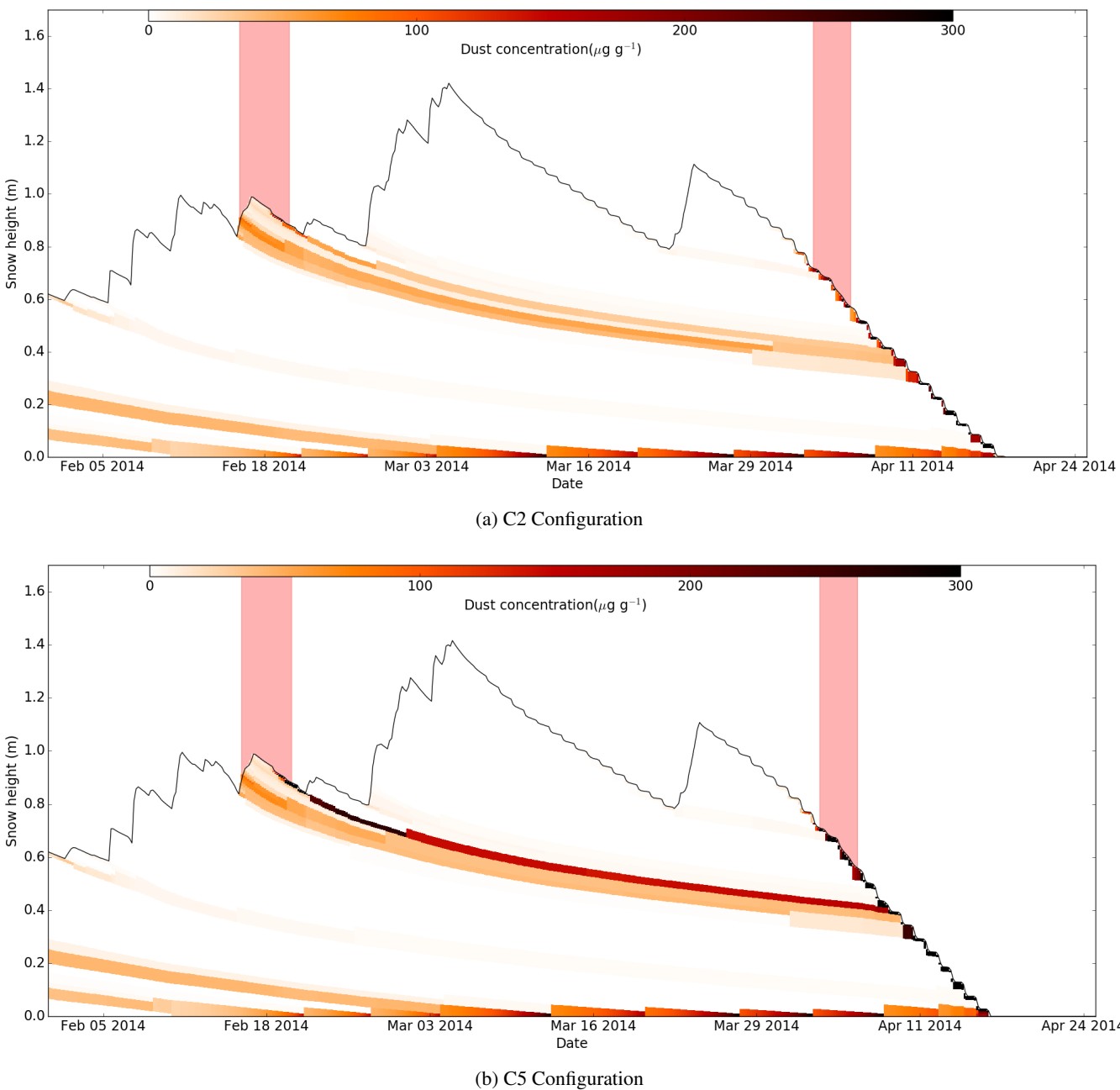

**Figure 8.** Simulated dust concentration profile for the second half of 2013/2014 snow season at Col de Porte. Panel (a) shows the configuration C2 using ALADIN-Climate deposition fluxes. Panel (b) shows C5 configuration using the same parameters but ALADIN-Climate deposition fluxes has been modified to reproduce the measurements by Di Mauro et al. (2015). The two major Saharan dust events are represented by the red areas.