# Peer review of "A multi-layer physically-based snowpack model simulating direct and indirect radiative impacts of light-absorbing impurities in snow"

_The Cryosphere, 2017_

## Short Comment (SC1) · 6 Aug 2017

Tuzet et al. use a sophisticated model to investigate the direct and indirect effects of light absorbing impurities on the melt of snow. The conclusion that the direct effect dominates over the season is expected, but it is interesting to see it demonstrated and quantified. I have some minor corrections and suggestion.

page 3

Explain briefly why radiative forcing increases as SA decreases.

page 5

It is not correct that LAI deposition fluxes measured in the field are used in this study.

page 7

Equation (2) seems to use subscript $i$ twice for different purposes: $D_i$ for deposition of impurity type i as in equation (1), and $z_i$ for layer i. $\Delta z_i$ is missing from the numerator.

"Each layer is affected the depth value of its center" is unclear.

$M_i$ and $SWE_i$ in equation (3) should be $M_o$ and $SWE_o$.

Is impurity content really stored on the ground after the snowpack has melted, and not just discarded by the model?

page 8

Equation (4) should really have subscripts for both impurity type and layer.

page 9

Is there a reference for ATMOTARTES?

What difference would also considering low cloud make?

Explain what SBDART is.

page 11

It is not correct to say that C5 is not included in the model evaluation; it can be seen in Table 2 and Figures 3, 4, 5 and 7.

page 13

While pointing out that C1 has the largest RMSE for snow depth, it should be noted that it has the smallest bias (and both the smallest bias and RMSE for SWE).

Why is the size of the bias between manual and automatic SWE measurements so

large? Morin et al. (2012) stated that the instrument is calibrated to manual measurements.

page 15

Transport of BC from Grenoble to Col de Porte could be supressed by persistent winter inversions.

Rather than using remote observations of dust in snow for the February event and none for the April event, why not scale ALADIN-Climate deposition in C5 to be closer to local BC equivalent measurements?

page 16

Albedo measurements are available at Col de Porte and could be compared with the simulations.

Figure 3 contradicts the assertion that C2, C3 and C4 improve the simulation at the end of the season compared to C1.

Table 2

The 20% scavenging is in the wrong column for C4

Figure 3

Why are the configuration lines broken in the upper panel and solid in the lower?

---

## Referee Comment (RC1) · Anonymous Referee #1 · 10 Aug 2017

———

General comments:

This paper introduces the updated detailed snowpack model Crocus, which now calculates the deposition and the evolution of light-absorbing impurities (LAI) such as black carbon (BC) and dust in the snowpack. Although the previous version of Crocus that incorporated the TARTES radiative transfer model can consider effects of SSA (specific surface area of snow) and LAI on snow albedo explicitly, the present update allows

model users of Crocus to simulate more realistic energy exchanges between the atmosphere and the snowpack as well as temporal evolution of snow physical conditions.

Overall, this paper is well written and I found there is potential that the present study can provide deepened knowledges of snow modelling; however, model validation works are not sufficient to demonstrate effectiveness of the present update. Model performances in terms of snow depth and snow water equivalent are almost the same between the present updated version and the reference version that calculates snow albedo by a relatively simple empirical approach. Therefore, I think readers will find it difficult to assess whether the present update successfully worked or not. At least, I think the authors should present model performance in terms of shortwave (broadband) albedo at Col de Porte in the same manner as Table 2.

——

Specific comments:

P6 L30- P7 L2: Is there a reference paper for the description of "The parameterization implemented in Crocus considers that the dry deposition affects the near-surface with an exponential decay to take into account wind pumping which buries a fraction of the dry deposited particles by circulating air into the uppermost snow layers."? An observation-based evidence for this description would be needed.

P8 L14-16: The authors state that "In the present study, the default value of BC scavenging coefficient is set to 20% according to the values provided in Flanner et al. (2007) and assessed by Doherty et al. (2013) and Yang et al. (2015)."; however, BC scavenging ratios listed in Table 1 (note that scavenging ratios for BC and dust listed in the table are inverted) are set to 0 % for most of the settings. Please explain why.

P10 L4: Lateral boundary forcing of meteorological conditions of the ALADIN-Climate model is given from ERA-Interim. How about lateral boundary forcing for BC and dust? In case an emission inventory is used in the parent model (boundary forcing), it should

be mentioned here as well.

P10-11 Sect. 3.3: The ALADIN-Climate-calculated LAI deposition fluxes were checked by referring to in-situ measurements obtained at Italian Alps. I think the authors should also check validity of the ALADIN-Climate-simulated precipitation rate at Col de Porte. This validation would reveal whether the ALADIN-Climate model could simulate wet deposition realistically or not.

P12 Sect. 4: Please add a subsection where validation results for shortwave albedo at Col de Porte are presented as mentioned above.

P14 L3-8: During the period when simulated near surface SSA are increased (new snow exists near the snow surface), observation data for SSA are not available as seen in the lower panel of Fig. 4. The authors should explain the reason.

P14 L21-22: When discussing radiative forcings due to direct and indirect impacts quantitatively, I think it is better to use C5 configuration as a control run rather than using C2 configuration. It is because C5 configuration gives more realistic LAI deposition fluxes, and values for radiative forcings would become more reliable and meaningful.

——

Technical corrections:

P7 L7: When introducing zj and j, please explain the coordinate system considered by Crocus (e.g., positive direction).

P7 L21: "Mo" and "SWE)o" are typos.

Figure 1: Please explain definitions for red and black circles explicitly.

---

## Referee Comment (RC2) · Anonymous Referee #2 · 11 Aug 2017

Tuzet et al., 2017 describe a state-of-the-art model suite to describe the evolution of a snow pack (snow accumulation, metamorphism and melt), with strongly improved capabilities to account for the impact of light absorbing impurities (LAI), namely black carbon (BC) and mineral dust. The snowpack model SURFEX/ISBA-Crocus is coupled to computation of in-snow radiative transfer (RT) with the model TARTES and atmospheric RT with ATMOTARTES, while deposition of LAI is simulated with the atmospheric model ALADIN-Climate. Comparing Crocus runs with and without accounting for the presence of LAI, the direct (snow darkening) and indirect (accelerated snow

grain metamorphism) of LAI are apportioned.

The paper presents a novel physically based approach to estimate the impact of LAI on snow albedo.

Two small points: Page 9 – subpoint 2.3: The atmospheric RT representation used by Tuzet et al., 2017 does not detailedly account for light absorbing aerosol and could be extended.

Page 1 – Abstract: Some of the formulations/statements in the paper in review should be improved or clarified (improper English language; like 14ff). What do you want to say with: Indeed, the model performances are not deteriorated compared to our reference Crocus version, while explicitly representing the impact of light-absorbing impurities.

---

## Short Comment (SC2) · 20 Aug 2017

The authors developed a sophisticated snowpack model to quantify radiative effects of LAIs in snow, which could potentially improve our understanding on aerosol contamination in snow. I have a few suggestions regarding two key factors in impurity-snow interactions, which may improve the discussions in the manuscript.

1. The authors assumed external mixing between LAIs and nonspherical snow grains using AART theory. However, recent studies (Liou et al., 2014; He et al., 2014) pointed

out that both impurity-snow internal mixing and snow nonsphericity play very important roles in snow albedo calculations. They showed that impurity-snow internal mixing can significantly enhances BC-induced snow albedo reduction compared with external mixing, but the enhancement is stronger for nonspherical snow grains than snow spheres, although spherical grains still have a larger absolute albedo reduction than nonspherical grains under the same BC content in snow. Thus, it is important to account for the combined effects of both key factors. I would recommend the authors to include these recent studies and add some discussions on this aspect.

2. Another important factor the authors did not mention is the underlying assumption of independent scattering among snow grains. However, snow is a close-packed medium in reality. He et al. (2017) recently found that snow close packing can reduce the albedo of pure snow by ∼0.01 at visible wavelengths and by up to ∼0.05 at near-infrared wavelengths, with even larger effects on dirty snow. Thus, it would be very helpful if the authors could include some discussions on this aspect.

References:

He, C., Li, Q. B., Liou, K. N., Takano, Y., Gu, Y., Qi, L., Mao, Y. H., and Leung, L. R.: Black carbon radiative forcing over the Tibetan Plateau, Geophys. Res. Lett., 41, 7806–7813, doi:10.1002/2014gl062191, 2014.

He, C., Y. Takano, and K. N. Liou: Close packing effects on clean and dirty snow albedo and associated climatic implications, Geophys. Res. Lett., 44, doi:10.1002/2017GL072916, 2017.

Liou, K. N., Takano, Y., He, C., Yang, P., Leung, L. R., Gu, Y., and Lee, W. L.: Stochastic parameterization for light absorption by internally mixed BC/dust in snow grains for application to climate models, J. Geophys. Res.-Atmos., 119, 7616–7632, doi:10.1002/2014jd021665, 2014.

---

## Referee Comment (RC3) · Anonymous Referee #3 · 22 Aug 2017

General comments:

the paper by Tuzet et al. proposes a very interesting integration of a snow model (CROCUS) with a radiative transfer model (TARTES) to estimate the impact of LAIs on the snow pack evolution in the French Alps. The authors calculate the direct and indirect radiative forcing and come up with an estimated earlier snow melt of about one week in 2014. The paper is well written and the messages are clear, it represents definitivey an advance in the study of LAIs on snow in Europe. There are only some issues to be resolved before final publication in TC.

[Figure]

I was quite impressed by the high concentration of BC estimated by the authors. In Figure 4, points represent the BC concentration estimated from measured spectral albedo (Dumont et al. 2017). I suggest to explicit it in the legend, otherwise the reader may think that they are the actual measured concentration of BC. To me, these concentrations are very high (more than 10^3 ppb), for example Khan et al. 2017 found similar values next to an active coal mine in the Arctic. A possible BC overestimation may lead to erroneous conclusions on the impact on snowpack dynamics. To present these data, the authors should validate the BC estimation from spectra, showing a quantitative correlation between estimated and measured BC concentration at Col de Porte. The only comparison provided regards the snow profile from 11 February 2014 (which is before the two dust events). From these plots, it is clear that the model is strongly overestimating the BC concentration (and underestimating dust). From this plot one may conclude that there is very little BC in Col de Porte. Furthermore, since both BC and MD impact the albedo in visible wavelengths, decoupling their effect from spectral data is still an open issue in the remote sensing of LAIs in snow (see for example Warren 2013 JGR). In my opinion, the estimation of BC from (hyper)spectral data should be always coupled with a validation scheme. The problem here may be hidden also in the spatial scale (as ackowledged in Section 5.1). ALADIN-climate works on a very coarse scale (50km) and the AWS used for this study provide a point measurement. It is understandable that the match is not perfect in simulated variables, but since the paper is focused on the impact of LAIs on snowpack evolution, I would ask: there was any BC in/on snow? If not, I would propose to strongly cut the discussion on BC and postpone it to a future paper in which actual BC measurements are provided. Another question on BC: where does it come from? It is plausible that it comes all from air contamination in Grenoble? Is there any atmospheric inversion that leads to the accumulation of BC in the lower atmosphere? Is ALADIN-climate able to reproduce it?

In the discussion section, the authors state that snowmelt advances 6-9 days due to LAIs deposition. This was due to BC or dust? If they ran the CROCUS simulations separately for the two impurities, it should be possible to estimate the partition of the

impact. I would expect that most of the advanced snomelt was due to the two big Saharan events in February and April 2014. If this is not true, maybe the overestimation of surface BC concentration may lead to erroneous conclusions. From an environmental/climate perspective it is very important to understand if some anthropogenic activity (e.g. BC emission from fossil fuel combustion) is involved in snow darkening in the European Alps.

——-

Specific comments:

pg3 line5: add some references here for the different type of impurities.

pg3 line26: actually the estimated advance was higher, please check the correct number in the referenced paper(s).

pg5 line12: replace "they" with "the author" (it was a single-author paper)

pg9 line22: replace "gaz" with "gas"

pg11 line11: please consider a reference to Varga et al. 2014, which also documents the Saharan events

pg17 line17: this is important, since Saharan dust particle diameter is usually 6-7microns. Assuming a Rayleigh scattering may lead to underestimate the impact of dust on snow. In any case, since you measured dust concentration with a Coulter counter, it would be useful to provide the measured mean diameter of dust particles from the profile of 11 February.

pg 19 line1: this is very interesting, last year a report was published in the journal "Neve e Valanghe" on this topic. You can find it here (http://www.aineva.it/pubblica/neve88/nv88_5.pdf), unfortunately it is available only in italian.

——

References:

Khan, A. L., H. Dierssen, J. P. Schwarz, C. Schmitt, A. Chlus, M. Hermanson, T. H. Painter, and D. M. McKnight (2017), Impacts of coal dust from an active mine on the spectral reflectance of Arctic surface snow in Svalbard, Norway, J. Geophys. Res. Atmos., 122, 1767–1778, doi:10.1002/2016JD025757.

Varga, G., Cserháti, C., Kovács, J., Szeberényi, J. and Bradák, B.: Unusual Saharan dust events in the Carpathian Basin (Central Europe) in 2013 and early 2014, Weather, 69(11), 309–313, doi:10.1002/wea.2334, 2014.

Warren, S. G. (2013). Can black carbon in snow be detected by remote sensing? Journal of Geophysical Research: Atmospheres, 118(2), 779–786. doi:10.1029/2012JD018476

---

## Author Comment (AC1) · 29 Sep 2017

**Response to SC1 by Richard Essery :**

a) Tuzet et al. use a sophisticated model to investigate the direct and indirect effects of light absorbing impurities on the melt of snow. The conclusion that the direct effect dominates over the season is expected, but it is interesting to see it demonstrated and quantified. I have some minor corrections and suggestion.

The authors are grateful to the referee for the positive global feedback on the work presented in the manuscript. The comments and additional grammar corrections have been helpful to improve the manuscript and are addressed point to point hereafter.

b) Page 3, explain briefly why radiative forcing increases as SSA decreases.

The radiative forcing increases as SSA decreases because the SSA decrease induces a decrease in the NIR reflectance of snow. This is due to the fact that a lower SSA is associated to a lower surface to mass ratio and, thus, a lower ratio between scattering and absorption.

Line 16 page 3 was then modified as follows: '… First, snow albedo in the near-infrared decreases with SSA (even in absence of LAI due to a decrease in the ratio between scattering and absorption coefficients ; e.g. Warren; 1982). '

c) page 5, It is not correct that LAI deposition fluxes measured in the field are used in this study.

Indeed, measured deposition fluxes are not used. The following corrections have been done:

Page 5 Lines 31-33 : In this study, the Crocus model takes typical meteorological driving data required for land surface models measured in the field, complemented by time series of LAI deposition fluxes (BC and dust) extracted from simulations with the ALADIN-Climate atmospheric model (Nabat et al. 2015). Our recent developments on the Crocus model were evaluated for the snow season 2013-2014 at the Col de Porte experimental site (Morin et al. 2012).

d) Page 7, Equation (2) seems to use subscript i twice for different purposes: $D_i$ for deposition of impurity type i as in equation (1), and $z_i$ for layer i. zi is missing from the numerator. (Why?)

In the revised manuscript, the subscript 'i' is used for impurity type and 'l' and 'k' for the layers. The changes are enlightened in all the equations of the revised manuscript (pages 6 to 8).

e) Each layer is affected the depth value of its center" is unclear.

Page 7 lines 6-8 have been modified has follows: 'Here $z_l$ is the depth of the layer l and $z_k$ is the depth of the layer k, N being the total number of Crocus layers. We assume the depth value of a layer to be the distance between the snowpack surface and the middle of this layer.'

f) Mi and SWEi in equation (3) should be Mo and SWEo.

Section 2.1.2 page 7 has been modified accordingly.

g) Is impurity content really stored on the ground after the snowpack has melted, and not just discarded by the model?

In this study, the impurity content of the basal layer is discarded when it melts.

Page 7 Line 25 has been modified accordingly : ' If the disappearing layer is the basal one, its impurity content is discarded by the model'.

h) Page 8 Equation (4) should really have subscripts for both impurity type and layer.

Done, please refer to the response to comment d).

i) Page 9

Is there a reference for ATMOTARTES?

There is no reference for ATMOTARTES. This manuscript is the first reference to it.

What difference would also considering low cloud make?

Since ATMOTARTES is only used to compute the spectral distribution of the solar irradiance, the difference between low clouds and high clouds would not significantly impact the results in terms of snow evolution.

Explain what SBDART is.

p9 line 23 has been modified accordingly: '… winter profiles from SBDART (Santa Barbara DISORT Atmospherice Radiatiave Transfer - Richiazzi et al., 1998). SBDART is a plane-parallel radiative transfer model for the atmosphere under clear and cloudy conditions. The solution of the radiative transfer equation is based on DISORT, so is more sophisticated and time consuming than the two flux method used in ATMOTARTES. '

j) page 11, It is not correct to say that C5 is not included in the model evaluation; it can be seen in Table 2 and Figures 3, 4, 5 and 7.

Indeed, C5 is included in the model evaluation.

The corresponding sentence  (Page 11 Line 20 ) has been removed.

k) Page 13,  While pointing out that C1 has the largest RMSE for snow depth, it should be noted that it has the smallest bias (and both the smallest bias and RMSE for SWE).

Page 13 Line 1 has been modified accordingly:  Over this period, the maximum RMSE is 8.0 cm (C1). It is to note that C1 has also the smallest bias because the underestimation of snow depth along the season (similar to all the other configurations) is compensated by a large overestimation of snow depth from May 20 onward.

Page 13 Line 12  The seasonal RMSE between measured and simulated SWE is 90.2 kg m$^{-2}$ for C0 and around 80.0 kg m$^{-2}$ for the other configurations. The minimum RMSE (71.6 kg m$^{-2}$) and bias (64.2 kg m$^{-2}$) are obtained for C1 configuration.

l) Page 13, Why is the size of the bias between manual and automatic SWE measurements so large? Morin et al. (2012) stated that the instrument is calibrated to manual measurements.

The automatic SWE measurement is calibrated using the weekly SWE manual measurement sites located immediately close to this instrument (SWE_North, SWE_South, see Morin et al., 2012). Here SWE measurements from the snowpit SWE measurement site are also used, exhibiting systematic deviations to the SWE measurements performed near the automatic SWE measurement site. Snow depth measurements are located at a third location, more or less in-between the SWE automatic sensor and the snowpit sensor.

m) Page 15, Transport of BC from Grenoble to Col de Porte could be suppressed by persistent winter inversions.

Small scale winter inversions (frequently observed in Grenoble) could indeed prevent BC transport from Grenoble to Col de Porte. This might be an explanation for the BC deposition overestimation by ALADIN-Climate because this model can not represent this small-scale phenomenon. The authors are grateful for this hypothesis, which has been added to the discussion Page 15 Line 15 :

Moreover persistent winter inversions are frequently observed in Grenoble. These phenomena could lead to accumulation of  BC emissions in the lower part of the atmosphere, preventing significant transport to Col de Porte. ALADIN Climate can not represent these winter inversions because of their relative small-scale compared to the model resolution. This may also partly explain the overestimation of BC deposition fluxes predicted by the model.

n) Rather than using remote observations of dust in snow for the February event and none for the April event, why not scale ALADIN-Climate deposition in C5 to be closer to local BC equivalent measurements?

Using deposition values scaled to reproduce the measurements would lead to unrealistic dust contents and could mask some model limitations. Indeed it is currently not possible to state whether there is not enough dust in the snowpack simulation or if dust impact is overestimated by the model because of modeling uncertainties. For these reasons, we decided to use realistic values found in the literature.

However, scaling ALADIN-Climate to be closer to local BC equivalent measurements is an interesting approach as well because it makes it possible to evaluate the performance of the model forced with the "optically correct" amount of impurities. An additional simulation has been performed to better reproduce BC equivalent measurements. Smaller RMSE/bias in terms of SSA and of shortwave albedo are observed (the albedo bias is reduced to 0.049 ) but the results in terms of snowdepth and SWE are deteriorated. Possible explanations for this deterioration and subsequent modifications in the manuscript are discussed in  response to the comment f) of RC1.

o) page 16 Albedo measurements are available at Col de Porte and could be compared with the simulations.

The evaluation of the new developments using albedo measurements has been added to the revised manuscript. Please refer to the response to the comment f) of RC1 for more details.

p) Figure 3 contradicts the assertion that C2, C3 and C4 improve the simulation at the end of the season compared to C1.

It is true that the assertion is valid only for snow depth and melting rate and not for SWE.

Page 16 – line 10 has been modified accordingly : "The atmospheric deposition fluxes provided by ALADIN-Climate (C2,C3 and C4) improve melting rate at the end of the season compared to C1 simulation although SWE is simulated more accurately using C1, probably due to a bias at the beginning of the season".

q) Table 2, The 20% scavenging is in the wrong column for C4

The mistake has been corrected in Table 2.

r) Figure 3, Why are the configuration lines broken in the upper panel and solid in the lower?

It was a mistake, the configuration lines are now broken for both panels for mote readability.

**Additionnal grammar corrections :**

**page 1**

referred **to as** Crocus

10
**the** Col de Porte experimental site

14
The model simulates snowpack evolution **reasonably**

15
**comma deleted**

16
from **the** ALADIN-Climate model

18
advances **by** 6 to 9 days

**page 3**

12
Lais radiative impact on snow → **the radiative impact of LAIs on snow**

15
accelerating near-surface SSA decrease

20
gathered information

21
Lais radiative impact  → the radiative impact of LAIs

22
absorption by LAIs

23
LAI content

34
referred **to as** dust outbreaks

**page 4**

drop significant amounts

the vertical profile of snowpack impurity content

LAI impacts

**page 11**

in mid-February

struck the Alps

**11**
the Italian Alps

**19**
The C5 simulation

**28**
In this way

**page 12**

**27**
Once this initial snowpack has melted

**page 13**

advances by 6 to 9 days

**page 14**

where → when

for the configurations implementing LAIs
almost the same

**17**
periods during which SWE is less than 50 kg m-1

**18**
increases through the season

**page 15**

which cannot represent

but small scale phenomena

**9**
affected by high levels

**11**
Grenoble**'s** impact

simulates LAI acceptably

in the presence

9
by 6 to 9 days

11
in this particular season

18
at different sites

22
LAI impacts

26
**the** Col de Porte experimental site

31
nutrient evolution

33
Crocus is now capable of tracking thin layers ... and representing the discontinuity induced in terms of

**References:**

Morin, S., Lejeune, Y., Lesaffre, B., Panel, J.-M., Poncet, D., David, P., and Sudul, M.: A 18-years long (1993 - 2011) snow and meteorological dataset from a mid-altitude mountain site (Col de Porte, France, 1325 m alt.) for driving and evaluating snowpack models, Earth Syst. Sci. Data, 4, 13–21, doi:10.5194/essd-4-13-2012, 2012.

Nabat, P., Somot, S., Mallet, M., Michou, M., Sevault, F., Driouech, F., Meloni, D., di Sarra, A., Di Biagio, C., Formenti, P., Sicard, M., Léon, J.- F., and Bouin,M.-N.: Dust aerosol radiative effects during summer 2012 simulated with a coupled regional aerosol–atmosphere–ocean model over the Mediterranean, Atmospheric Chemistry and Physics, 15, 3303–3326, doi:10.5194/acp-15-3303-2015, http://www.atmos-chem-phys.net/15/3303/2015/, 2015.

Ricchiazzi, P., Yang, S., Gautier, C., and Sowle, D.: SBDART: A research and teaching software tool for plane-parallel radiative transfer in the 35 Earth's atmosphere., Bull. Am. Met. Soc., 79, 2101–2114, 1998.

---

## Author Comment (AC2) · 29 Sep 2017

**Response to RC1 :**

**General comments:**

a)  This paper introduces the updated detailed snowpack model Crocus, which now calculates the deposition and the evolution of light-absorbing impurities (LAI) such as black carbon (BC) and dust in the snowpack. Although the previous version of Crocus that incorporated the TARTES radiative transfer model can consider effects of SSA (specific surface area of snow) and LAI on snow albedo explicitly, the present update allows model users of Crocus to simulate more realistic energy exchanges between the atmosphere and the snowpack as well as temporal evolution of snow physical conditions. Overall, this paper is well written and I found there is potential that the present study can provide deepened knowledges of snow modelling; however, model validation works are not sufficient to demonstrate effectiveness of the present update. Model performances in terms of snow depth and snow water equivalent are almost the same between the present updated version and the reference version that calculates snow albedo by a relatively simple empirical approach. Therefore, I think readers will find it difficult to assess whether the present update successfully worked or not. At least, I think the authors should present model performance in terms of shortwave (broadband) albedo at Col de Porte in the same manner as Table 2.
* * *
The authors are grateful to the reviewer for reviewing our manuscript and for the suggestions concerning the model validation. Indeed there are no real improvement in terms of snow water equivalent and depth but it is important to keep in mind that the "relatively simple empirical approach" used in the reference Crocus version was calibrated at Col de Porte. This simple approach is consequently expected to give satisfying results at Col de Porte and significant improvements were not expected there by improving the physics of the snow model, given that the performance of the model is already virtually as good as it can be measured, given all the uncertainties at play (meteorological observations, snow measurements, model errors – see Lafaysse et al., 2017). We are satisfied that the more sophisticated model has similar performance than the default version. This is discussed in more detailed p16 in the revised manuscript.
However, the empirical scheme of snow darkening used in the reference version can not be applied as such to areas where LAIs contamination levels are significantly different from Col de Porte (or the parameterization should be manually adjusted otherwise spurious results are obtained, see e.g. Jacobi et al., 2015 or 2016). The new scheme using LAI deposition fluxes as inputs of the model is expected to be more transferable to other sites, as long as appropriate deposition fluxes are available. Moreover, the recent developments make it possible to numerically investigate LAI-snow interaction processes.

The evaluation of daily shortwave albedo has been added as detailed in response to comment f).

**Specific comments:**

b) P6 L30- P7 L2: Is there a reference paper for the description of "The parameterization implemented in Crocus considers that the dry deposition affects the near-surface with an exponential decay to take into account wind pumping which buries a fraction of the dry deposited particles by circulating air into the uppermost snow layers."? An observation-based evidence for this description would be needed.

The authors consider that wind pumping might be a process affecting the redistribution of dry-deposed LAIs in the near-surface snowpack. However we have no observation-based evidence to provide in support of this  intuition. Hence, we used a low value for the e-folding depth of the dry deposition distribution (5 mm) providing similar LAI distribution than affecting all the deposition to the topmost layer (basic parameterization of dry deposition) as explained in the manuscript p 7L10.

In addition, as detailed in the paper, the value is in accordance with experimentally measured depth for which wind pumping has an effect.

c) P8 L14-16: The authors state that "In the present study, the default value of BC scavenging coefficient is set to 20% according to the values provided in Flanner et al. (2007) and assessed by Doherty et al. (2013) and Yang et al. (2015)."; however, BC scavenging ratios listed in Table 1 (note that scavenging ratios for BC and dust listed in the table are inverted) are set to 0 % for most of the settings. Please explain why.

The default value of BC scavenging in our study is set to 0% with just one configuration implementing the BC scavenging value of 20% provided in Flanner et al. (2007). The corresponding paragraph has been modified.

The mistake in the table has been corrected.

The legend of Figure 2 has also been modified accordingly (p27):  Simulated BC concentration evolution at the end of 2013/2014 snow season at Col de Porte. The upper panel corresponds to a simulation without scavenging  whereas the lower panel corresponds to a simulation using the value of 20% for BC scavenging.

Page 8 Lines 14-16 have been modified accordingly: In the present study, we disabled scavenging by default, implying that the default value of BC scavenging coefficient is set to 0%. However in order to assess the impact of BC scavenging we run a configuration implementing a BC scavenging coefficient of 20% according to the values provided in Flanner et al. (2007) and assessed by Doherty et al. (2013) and Yang et al. (2015).

d) P10 L4: Lateral boundary forcing of meteorological conditions of the ALADIN-Climate model is given from ERA-Interim. How about lateral boundary forcing for BC and dust? In case an emission inventory is used in the parent model (boundary forcing), it should be mentioned here as well.

P10 L4 has been modified accordingly: For aerosols, no data are available at the lateral boundaries. Aerosol lateral boundary forcing is set to 0 because ALADIN-Climate domain is considered to be large enough to include all the aerosol sources affecting the area. For instance, the domain includes the whole Saharan desert.

e) P10-11 Sect. 3.3: The ALADIN-Climate-calculated LAI deposition fluxes were checked by referring to in-situ measurements obtained at Italian Alps. I think the authors should also check validity of the ALADIN-Climate-simulated precipitation rate at Col de Porte. This validation would reveal whether the ALADIN-Climate model could simulate wet deposition realistically or not.

 In this study, the precipitation rate comes from in-situ measurements at Col de Porte. The wet deposition fluxes from ALADIN-Climate are only activated if there is in-situ measured precipitation. Dry deposition is active all the time. These details were missing in the first version of the manuscript,  and are now added to the revised manuscript.

Page 10- line 7 has been modified : '...(Di Mauro et al., 2015). Wet deposition is only activated when there is measured precipitation. '

Modifications have been performed in the discussion : page 15 – line 29 :
"Lastly, it must be underlined that the wet deposition fluxes from ALADIN-Climate are only taken into account in the simulations when in-situ precipitation is measured. Consequently, any mismatch between ALADIN-Climate and measured precipitation occurrence may lead to errors in simulated wet deposition. "

f) P12 Sect. 4: Please add a subsection where validation results for shortwave albedo at Col de Porte are presented as mentioned above.

Additional evaluations were performed to address this suggestion.

Albedo measurements are available from two sensors at Col de Porte: daily broadband albedo described in Morin et al., 2012 since 1993 and spectral albedo measurements for the snow season 2013-2014 described in Dumont et al., 2017.

First, the simulated daily broadband albedo was evaluated using broadband albedo calculated from daily averaged downwelling and upwelling shortwave broadband radiation fluxes, hourly measured at Col de Porte. Measurements were discarded during snowfall events or when measured fluxes are too low: lower than 20 W m$^{-2}$ for the incoming radiation and than 2 W m$^{-2}$ for the reflected radiation (Lafaysse et al. 2017, Morin et al. 2012). If less than 5 hourly data can be used for calculation daily albedo were discarded .

The daily broadband albedo was computed using model results for each configuration (discarding the same data as for measurements). The results presented a significant bias of around -0.1 (Figure and table bellow).

[Figure]

Figure 1:  Daily broadband albedo measured and simulated with our different model configurations

| Configuration | Shortwave albedo |
|---|---|
| | RMSE(bias) from 05/11/13 to 01/05/14 |
| C0 | 0.100(-0.081) |
| C1 | 0.144(-0.112) |
| C2 | 0.110(-0.075) |
| C3 | 0.113(-0.078) |
| C4 | 0.111(-0.077) |
| C5 | 0.106(-0.072) |

Table 1: RMSE and bias between measured and simulated daily broadband albedo

A similar bias between daily albedo and broadband albedo derived from spectral measurements (Dumont et al., 2017) was noticed in Lafaysse et al. 2017 (Figure 1). A possible explanation for this systematic bias is the slope of the snow surface under the sensor.

Secondly, the evaluation was thus restricted to broadband albedo values derived from spectral measurements (Dumont et al., 2017). These values have indeed been corrected from slope effect and a value of broaband albedo of a perfectly flat surface can be derived. The evaluation with respect to this dataset has been added to the manuscript as detailed in the following.

Data & Methods section : P10 L1

[revised manuscript text omitted]

g) P14 L3-8: During the period when simulated near surface SSA are increased (new snow exists near the snow surface), observation data for SSA are not available as seen in the lower panel of Fig. 4. The authors should explain the reason.

Near surface SSA are obtained via spectral albedo measurements. These measurements are less accurate or unavailable in case of snow falls as detailed in Dumont et al., 2017.
The following sentence was added page 10 line 22 :
"Near surface LAI content and SSA are generally not available during snowfall due to large uncertainties in the albedo measurement (Dumont et al., 2017). "

h) P14 L21-22: When discussing radiative forcings due to direct and indirect impacts quantitatively, I think it is better to use C5 configuration as a control run rather than using C2 configuration. It is because C5 configuration gives more realistic LAI deposition fluxes, and values for radiative forcings would become more reliable and meaningful.

In order to address this suggestion, the same method has been applied to C5 configuration. It appears that using C5 as a control run leads to the same temporal patterns as described in the manuscript discussion (Figure 3 bellow). However the distribution between direct and indirect impacts is slightly modified, with 14.1% of indirect impact against 15.3%. For the present study the default control run (C2) has not been changed but the results obtained using C5 as a control run are mentionned.

[Figure]

Figure 3: Same figure as Figure 6 in the original manuscript but using C5 as a control run instead of C2.
Energy absorbed by the snowpack during the season (upper panel); the full lines correspond to the daily amount of energy absorbed whereas the dashed lines corresponds to the cumulative energy absorbed over the study period. $R_{ind,daily}$ compared to near-surface SSA computed from C1 (lower panel); $R_{ind,daily}$ is the daily relative importance of LAIs in snow radiative forcing coming from the indirect impact. The dates during which the ground influences the energy budget have been masked (grey shading). The red shading represents two major Saharan dust events.

A note has been added P12 L13: Note that the same method can be applied by replacing C2 with C3,C4 or C5.

A paragraph on this additional result has been added in Section 4.5 Page 14 Line 27: Sections 4.2 and 4.4 highlight that C5 provides better results than C2 in terms of near-surface LAIs concentration and shortwave albedo. Given that radiative forcing is expected to be more accurate for C5, the same method has also been applied using C5 as a control run (instead of C2 on Figure 6). We obtain similar results in term of temporal evolution but the distribution between the average direct and indirect impacts is only slightly modified, with 14.1\% attributed to the indirect impact instead of 15.3 %, which we consider an insignificant variation.
——

**Technical corrections:**

i) P7 L7: When introducing zj and j, please explain the coordinate system considered by Crocus (e.g., positive direction).

Page 7 Line 7 has been modified accordingly: The layer number 1 is the topmost layer whereas the layer number N is the bottom layer

j) P7 L21: "Mo" and "SWE)o" are typos.

Done.

k) Figure 1: Please explain definitions for red and black circles explicitly.

[Figure]

The red circles represent mineral dust and the black circles represent black carbon. The definitions had not been put explicitly on the figure because the model can easily account for other types of LAIs. As the other types of LAIs are not accounted for in this study, the figure has been changed.

: Mineral dust

: Black carbon

Figure 4 (Modified in the manuscript): Description of the detailed snowpack model Crocus including an explicit representation of LAIs deposition and evolution.

---

## Author Comment (AC3) · 29 Sep 2017

**Response to RC2:**

a) Tuzet et al., 2017 describe a state-of-the-art model suite to describe the evolution of a snow pack (snow accumulation, metamorphism and melt), with strongly improved capabilities to account for the impact of light absorbing impurities (LAI), namely black carbon (BC) and mineral dust. The snowpack model SURFEX/ISBA-Crocus is coupled to computation of in-snow radiative transfer (RT) with the model TARTES and atmospheric RT with ATMOTARTES, while deposition of LAI is simulated with the atmospheric model ALADIN-Climate. Comparing Crocus runs with and without accounting for the presence of LAI, the direct (snow darkening) and indirect (accelerated snow grain metamorphism) of LAI are apportioned.
The paper presents a novel physically based approach to estimate the impact of LAI on snow albedo.

The author are grateful for the review and positive feedbacks that help improving the manuscript. A response to each comments is provided hereafter.

**Two small points:**

b) Page 9 – subpoint 2.3: The atmospheric RT representation used by Tuzet et al., 2017 does not detailedly account for light absorbing aerosol and could be extended.

The atmospheric model indeed only account for one type of aerosols with a fixed vertical profile. It could be extended. We however think that the impact of such improvement would be small since the model is only used to compute the spectral distribution of the irradiance.
A statement about this has been added in the discussion (page 17 line 14):
Concerning atmospheric radiative transfer (Section 2.3), ATMOTARTES only has a rough representation of the effect of LAIs in the atmosphere (one type of aerosols and constant vertical profile). This could be extended as in SBDART (Richiazzi et al., 1998) but the impact would be limited while the numerical cost would be significantly increased.

c) Page 1 – Abstract: Some of the formulations/statements in the paper in review should be improved or clarified (improper English language; like 14ff). What do you want to say with: Indeed, the model performances are not deteriorated compared to our reference Crocus version, while explicitly representing the impact of light-absorbing impurities.

The abstract was modified as follows :

Page 1 Line 13 has to be modified: The model simulates snowpack evolution reasonably, providing similar performances to our reference Crocus version in term of snow depth, snow water equivalent, near-surface specific surface area and shortwave albedo. Since the reference empirical albedo scheme was calibrated at Col de Porte, improvements were not expected to be significant in this study.

**References:**

Ricchiazzi, P., Yang, S., Gautier, C., and Sowle, D.: SBDART: A research and teaching software tool for plane-parallel radiative transfer in the 35 Earth's atmosphere., Bull. Am. Met. Soc., 79, 2101–2114, 1998.

---

## Author Comment (AC4) · 29 Sep 2017

**Response to SC2 by Cenlin He :**

a) The authors developed a sophisticated snowpack model to quantify radiative effects of LAIs in snow, which could potentially improve our understanding on aerosol contamination in snow. I have a few suggestions regarding two key factors in impurity-snow interactions, which may improve the discussions in the manuscript.

The authors are grateful to the referee for these suggestions on LAI-snow interactions, which enrich the discussion part of the manuscript.

b) 1. The authors assumed external mixing between LAIs and nonspherical snow grains using AART theory. However, recent studies (Liou et al., 2014; He et al., 2014) pointed out that both impurity-snow internal mixing and snow nonsphericity play very important roles in snow albedo calculations. They showed that impurity-snow internal mixing can significantly enhances BC-induced snow albedo reduction compared with external mixing, but the enhancement is stronger for nonspherical snow grains than snow spheres, although spherical grains still have a larger absolute albedo reduction than nonspherical grains under the same BC content in snow. Thus, it is important to account for the combined effects of both key factors. I would recommend the authors to include these recent studies and add some discussions on this aspect.

Page 17 Line 20 has been modified accordingly: Finally, in the present study LAIs are assumed to be externally mixed to the ice matrix. Flanner et al. (2012) showed that internally mixed BC was up to 80\% more absorptive than externally mixed BC. Recently, Liou et al., 2014 and He et al., 2014 also pointed out that both impurity-snow internal mixing and snow nonsphericity play very important roles in snow albedo calculations. They showed that internal mixing can enhances BC-induced snow albedo reduction up to 50% compared with external mixing. This enhancement is stronger for nonspherical ice elements than ice spheres, although ice spheres still have a larger absolute albedo reduction than nonspherical ice elements under the same BC content in snow. Introducing an internally-mixed representation of LAIs in TARTES could in turn impact the results. However, a better knowledge of the partition between internally and externally mixed LAIs in seasonal snowpacks would be required to accurately characterize the impact of this variable.

c) 2. Another important factor the authors did not mention is the underlying assumption of independent scattering among snow grains. However, snow is a close-packed medium in reality. He et al. (2017) recently found that snow close packing can reduce the albedo of pure snow by 0.01 at visible wavelengths and by up to 0.05 at nearinfrared wavelengths, with even larger effects on dirty snow. Thus, it would be very helpful if the authors could include some discussions on this aspect.

The AART used in TARTES exploits the fact that, for large particles with respect to the wavelength and weakly-absorbing, the radiative transfer equation for dense media has the same form as the conventional (sparse medium) one, and that the free path length and absorption, which ultimately determines the macroscopic properties, are not affected by the concentration in the medium (e.g. Kokhanovsky 2004, chapter 4). This is an important result that support the validity of numerous works on albedo simulation with RT for snow (e.g. Warren and Wiscombe, 1980). It is true that scattering coefficient and phase function are affected by medium concentration; but both effects compensate each other owing to the similarity principle in the RT equation (C. Mitrescu, , G.L. Stephens, On similarity and scaling of the radiative transfer equation, Journal of Quantitative Spectroscopy and Radiative Transfer 86, 4, 387–394, 2004). This discussion is beyond the scope of the manuscript.

**References:**

Flanner, M., Liu, X., Zhou, C., and Penner, J.: Enhanced solar energy absorption by internally-mixed black carbon in snow grains, Atmos. Chem. Phys., 12, 4699–4721, doi:doi:10.5194/acp-12-4699-2012, 2012.

He, C., Li, Q. B., Liou, K. N., Takano, Y., Gu, Y., Qi, L., Mao, Y. H., and Leung, L. R.: Black carbon radiative forcing over the Tibetan Plateau, Geophys. Res. Lett., 41, 7806–7813, doi:10.1002/2014gl062191, 2014.

He, C., Y. Takano, and K. N. Liou: Close packing effects on clean and dirty snow albedo and associated climatic implications, Geophys. Res. Lett., 44, doi:10.1002/2017GL072916, 2017.

Kokhanovsky, Alex A. *Light scattering media optics*. Springer Science & Business Media, 2004.

Liou, K. N., Takano, Y., He, C., Yang, P., Leung, L. R., Gu, Y., and Lee, W. L.: Stochastic parameterization for light absorption by internally mixed BC/dust in snow grains for application to climate models, J. Geophys. Res.-Atmos., 119, 7616–7632, doi:10.1002/2014jd021665, 2014.

Warren, S. G. and Wiscombe, W.: A Model for the Spectral Albedo of Snow. II: Snow Containing Atmospheric Aerosols, J. Atmos. Sci., 37,2734–2745, 1980.

---

## Author Comment (AC5) · 29 Sep 2017

**Response to RC3:**

**General comments:**

a) the paper by Tuzet et al. proposes a very interesting integration of a snow model (CROCUS) with a radiative transfer model (TARTES) to estimate the impact of LAIs on the snow pack evolution in the French Alps. The authors calculate the direct and indirect radiative forcing and come up with an estimated earlier snow melt of about one week in 2014. The paper is well written and the messages are clear, it represents definitivey an advance in the study of LAIs on snow in Europe. There are only some issues to be resolved before final publication in TC.

The author are grateful to the referee for this review and interest in the manuscript, the issues highlighted are addressed in the point by point response hereafter.

b) I was quite impressed by the high concentration of BC estimated by the authors. In Figure 4, points represent the BC concentration estimated from measured spectral albedo (Dumont et al. 2017). I suggest to explicit it in the legend, otherwise the reader may think that they are the actual measured concentration of BC. To me, these concentrations are very high (more than 10ˆ3 ppb), for example Khan et al. 2017 found similar values next to an active coal mine in the Arctic.

The concentrations estimated from measured spectral albedo (Dumont et al. 2017) are BC equivalent concentrations. They include all type of LAIs such as mineral dust, organic debris, organic carbon. This could explain the high concentrations found. Flanner et al. (2007) report concentrations of BC in the Alps up to 800ng/g, so accounting for the other types of LAIs (especially mineral dust and organic debris) it is not unrealistic to have this BC equivalent concentration. Note also that there is also a high concentration of plant debris in Col de Porte snow due to the nearby forest.

The label "Measured" has been replaced by "based on measured albedo" as it was already for near-surface SSA. This is also explained page 10 lines 21-22 in the manuscript.

c) A possible BC overestimation may lead to erroneous conclusions on the impact on snowpack dynamics. To present these data, the authors should validate the BC estimation from spectra, showing a quantitative correlation between estimated and measured BC concentration at Col de Porte. The only comparison provided regards the snow profile from 11 February 2014 (which is before the two dust events). From these plots, it is clear that the model is strongly overestimating the BC concentration (and underestimating dust).

Possible sources of BC overestimation are discussed in section 5.1. However considering that the model simulates reasonably well the BC equivalent content (ie. meaning correct radiative forcing), we believe that even if BC is overestimated and dust underestimated, more accurate LAI simulated content will not improve the results in terms of snow melt rate. See also responses and modifications to the comment d) below.

d) From this plot one may conclude that there is very little BC in Col de Porte. Furthermore, since both BC and MD impact the albedo in visible wavelengths, decoupling their effect from spectral data is still an open issue in the remote sensing of LAIs in snow (see for example Warren 2013 JGR). In my opinion, the estimation of BC from (hyper)spectral data should be always coupled with a validation scheme.

Unfortunately, only one measurements of BC at Col de Porte has been performed this year. This issue is already discussed in Dumont et al., (2017). A discussion point has been added in the paper: page 15 –line 29
The upper panel of Figure 4 points out that C5 improves the simulated late season near-surface impurity concentrations compared to all other configurations.
However, in order to test this hypothesis a more detailed evaluation of the LAI (BC and dust) contents in snow should be performed using direct measurements of LAI and not LAI content estimated from (hyper)spectral measurements (e.g. Warren, 2013) which are uncertain for low impurity content (Dumont et al., 2017) but is beyond the scope of the present study. "

e) The problem here may be hidden also in the spatial scale (as ackowledged in Section 5.1). ALADIN-climate works on a very coarse scale (50km) and the AWS used for this study provide a point measurement. It is understandable that the match is not perfect in simulated variables, but since the paper is focused on the impact of LAIs on snowpack evolution, I would ask: there was any BC in/on snow? If not, I would propose to strongly cut the discussion on BC and postpone it to a future paper in which actual BC measurements are provided.

See responses to comments c) and d). The discussion on BC in snow has been kept in the revised version of the manuscript since it highlights the limitations of the modeling chain and of the evaluation dataset.

f) Another question on BC: where does it come from? It is plausible that it comes all from air contamination in Grenoble? Is there any atmospheric inversion that leads to the accumulation of BC in the lower atmosphere? Is ALADIN-climate able to reproduce it?

Winter atmospheric inversions are indeed commonly observed in Grenoble. Considering the coarse scale of ALADIN-Cimate, these events can not be represented correctly .
The response m) of the specific comment RC1 for a more detailed response and subsequent modification in the paper further addresses this topic.

g) In the discussion section, the authors state that snowmelt advances 6-9 days due to LAIs deposition. This was due to BC or dust? If they ran the CROCUS simulations separately for the two impurities, it should be possible to estimate the partition of the impact. I would expect that most of the advanced snowmelt was due to the two big Saharan events in February and April 2014.

In order to address this question, additional simulations with BC only or dust only have been performed. The results show that for C2, C3 and C4 BC is responsible for most of the radiative impact whereas for C5 half of the radiative impact originates from dust. However, since we are not able to accurately evaluate the simulated BC and dust contents separately (see responses to comments c) and d)), we decided not to include these results in the paper.

These limitations have been however underlined page 18 – line 19 : For example, a direct evaluation of the dust and BC contents is required to quantify more precisely their respective part in the shortening of the snow season.

h) If this is not true, maybe the overestimation of surface BC concentration may lead to erroneous conclusions. From an environmental/ climate perspective it is very important to understand if some anthropogenic activity (e.g. BC emission from fossil fuel combustion) is involved in snow darkening in the European Alps.

An overestimation of surface BC concentration may lead to an overestimation of the melt rate or may be compensated by an underestimation of the mineral dust concentration. We do not have enough chemical measurements at Col de Porte to accurately conclude on the partition between mineral dust and BC relative impacts. However if ALADIN-Climate deposition fluxes are correct, at least half of the impact comes from BC (cf response f) above).

**Specific comments:**

i) pg3 line5: add some references here for the different type of impurities.

References for the different types of impurities have been added.

Page 3 Line 5  has been modified accordingly: such as mineral dust (Painter et al. 2010), black carbon (BC) from combustion sources (Flanner et al. 2007),  volcanic ash (Conway et al. 1996), soil organics (Takeuchi 2002), algae, and other biological organisms and constituents (Cook et al. 2017)

j) pg3 line26: actually the estimated advance was higher, please check the correct number
in the referenced paper(s).

Painter et al. (2013) indeed pointed out that the shift in total melt-out due to dust radiative forcing can be up to 50 days.

The reference Page 3 Line26 has been modified accordingly: can advance total melt-out by up to 50 days

k) pg5 line12: replace "they" with "the author" (it was a single-author paper)

Done

l) pg9 line22: replace "gaz" with "gas"

Done

m) pg11 line11: please consider a reference to Varga et al. 2014, which also documents
the Saharan events

This has been included in the introduction
Page 4 Line 1: dust outbreaks,  are very sporadic events mostly occurring from April to August (Varga et al. 2014)

n) pg17 line17: this is important, since Saharan dust particle diameter is usually 6-
7microns. Assuming a Rayleigh scattering may lead to underestimate the impact of
dust on snow. In any case, since you measured dust concentration with a Coulter
counter, it would be useful to provide the measured mean diameter of dust particles
from the profile of 11 February.

The Coulter counter measurements indeed provide information on dust particles diameter. Assuming dust particles to be spheres, we calculate their volume and compute a volume-weighted size distribution of dust particles. Figure 5 below presents this size distribution of dust particles according to their volume contribution which has a mode around 3 micrometers .

[Figure]

Figure 5: Dust particles diameter distribution according to their volume contribution, obtained from the Coulter counter measurements performed on the 11 February 2013 at Col de Porte.

Page 17 Line 17 has been modified accordingly: This theory is acceptable in the case of BC but may not perfectly apply to dust, depending on its volume size distribution, and may lead to an underestimation of dust radiative impacts. Coulter measurements show that the average diameter according to their volume contribution for our dust is 2.8 µm, which indeed suggest that dust radiative impact can be over-estimated in this study.

o) pg 19 line1: this is very interesting, last year a report was published
in the journal "Neve e Valanghe" on this topic. You can find it here
(http://www.aineva.it/pubblica/neve88/nv88_5.pdf), unfortunately it is available only in
italian.

The authors are grateful for this reference, in the future the authors consider using the recent developments in Crocus to investigate the link between Saharan dust outbreaks and snow stability.

**References:**

Dumont, M., Arnaud, L., Picard, G., Libois, Q., Lejeune, Y., Nabat, P., Voisin, D., and Morin, S.: In situ continuous visible and near-infrared spectroscopy of an alpine snowpack, The Cryosphere, 11, 1091–1110, doi:10.5194/tc-11-1091-2017, http://www.the-cryosphere.net/11/ 1091/2017/, 2017.

Flanner, M. G., Zender, C. S., Randerson, J. T., and Rasch, P. J.: Present-day climate forcing and response from black carbon in snow, J.30 Geophys. Res., 112, D11 202, doi:10.1029/2006JD008003, 2007.

Khan, A. L., H. Dierssen, J. P. Schwarz, C. Schmitt, A. Chlus, M. Hermanson, T. H. Painter, and D. M. McKnight (2017), Impacts of coal dust from an active mine on the spectral reflectance of Arctic surface snow in Svalbard, Norway, J. Geophys. Res. Atmos., 122, 1767–1778, doi:10.1002/2016JD025757.

Painter, T. H., Seidel, F. C., Bryant, A. C., McKenzie Skiles, S., and Rittger, K.: Imaging spectroscopy of albedo and radiative forcing by 25 light-absorbing impurities in mountain snow, Journal of Geophysical Research: Atmospheres, 118, 9511–9523, 2013b.

Varga, G., Cserháti, C., Kovács, J., Szeberényi, J. and Bradák, B.: Unusual Saharan dust events in the Carpathian Basin (Central Europe) in 2013 and early 2014, Weather, 69(11), 309–313, doi:10.1002/wea.2334, 2014.

---

## Author Comment (AC6) · 29 Sep 2017

[revised manuscript text omitted]

10 Here,  $M_{t,l,i}$ and $M_{t+\delta t,l,i}$ represent the mass of impurity type $i$ in g m$^{-2}$ in the layer  $l$ at the beginning and end of the time step $\delta t$,  $D_i$ the dry deposition flux expressed in g m$^{-2}$ s$^{-1}$ and $h$ is the user-defined e-folding depth characterizing the decrease rate of the impurity distribution with depth.  Here $z_l$ is the depth of the layer l and $z_k$ is the depth of the  layer k, N being the total number of Crocus layers. We assume the depth value of a layer to be the distance

15 between the snowpack surface and the middle of this layer. The default value for $h$ is set to 5 mm according to the range of value in Clifton et al. (2008), which shows that wind-pumping affects between 1 and 10 mm of the snowpack surface depending on snow and atmospheric properties. As the typical thickness of the surface layer in Crocus is close to 1 cm, this value of $h$ implies that most of the  LAIs are initially deposited in the uppermost layer.

**2.1.2 LAI evolution within the snowpack**

20 **Handling of layers**

Crocus manages the layers to keep their number under a prescribed maximum value. When there are too many layers, two layers having similar microstructure properties can merge and the properties of the newly created layer are re-calculated (see details in Charrois et al., 2016 or Vionnet et al., 2012). Concerning LAIs content, the impurity mass of the new layer is the sum of the impurity mass of the two old layers.

25 On the contrary, when there are fewer layers than the optimum value computed by Crocus,  a thick layer ($t$) can be split into two different layers. For each of the newly created layers ($n$), the impurity mass is apportioned according to their snow water equivalent (SWE):

$$M_n = M_{it} \times \frac{\text{SWE}_n}{\text{SWE}_{it}}, \tag{3}$$

$M_n$ and $M_{\overline{o\,t}}$ being respectively the impurity mass of the newly created and the initial layers in g m$^{-2}$ and SWE$_n$ and SWE$_{)\overline{o\,t}}$ the SWE of the newly created and the initial  layer in kg m$^{-2}$.

If a snow layer completely disappears (e.g. due to total melt or sublimation), its impurity content is transferred to the layer below leading to an accumulation of LAI on the top of the snowpack during melt time. This enrichment process has been widely observed (e.g., Skiles, 2014; Yang et al., 2015). If the disappearing layer is the basal one, its impurity content is discarded by the model.

**Scavenging**

It has been established  that some LAI types can be partially scavenged with percolating water during melt time (e.g., Flanner et al., 2007; Doherty et al., 2013; Sterle et al., 2013; Yang et al., 2015). When liquid water percolates into the snowpack, it can carry part of its impurity mass to the layer below. In the current version of Crocus, water percolation is handled following a simple and conceptual bucket approach (Lafaysse et al., 2017). Each layer $(l)$ is seen as a homogeneous reservoir containing a given volumetric liquid water content $W_{\overline{liq}}W_{liq,l}$. For each layer a maximum volumetric liquid water holding capacity $W_{\overline{liqmax}}W_{liqmax,l}$ is defined as a percentage of the pore volume. If $W_{\overline{liq}}$ exceeds $W_{\overline{liqmax}}W_{liq,l}$ exceeds $W_{liqmax,l}$, the excess water $F_{\overline{liq}}F_{liq,l}$ drains to the underlying layer.

Similarly to Flanner et al. (2007), we assume LAI inclusion in melt water proportional to its mass mixing ratio multiplied by a scavenging factor. Therefore, a scavenging coefficient $C_{scav,i}$, adjustable for each impurity type $(i)$, has been introduced in the model. In case of water percolation, for each layer $(l)$ the scavenged mass $M_{\overline{scav,i}}M_{scav,i,l}$ is calculated with:

$$M_{\underline{scav,i}\,scav,i,l} = F_{\underline{liq}\,liq,l} \times C_{\underline{scav,i}\,
[revised manuscript text omitted]